# One model to rule them all: Unification of voltage-gated potassium channel models via deep non-linear mixed effects modelling

Domas Linkevicius[1,2], Angus Chadwick[1], Melanie I. Stefan[3], David C. Sterratt[1]*

**1** Institute for Machine Learning, School of Informatics, University of Edinburgh, Edinburgh, United Kingdom, **2** Computational Neuroscience Unit, Okinawa Institute of Science and Technology, Okinawa, Japan, **3** Faculty of Medicine, Medical School Berlin, Berlin, Germany

\* David.C.Sterratt@ed.ac.uk

## Abstract

Ion channels are essential for signal processing and propagation in neural cells. Voltage-gated ion channels permeable to potassium ($K_v$) form one of the most prominent channel families. Techniques used to model the voltage-dependent gating of $K_v$ channels date back to Hodgkin and Huxley (1952). Different $K_v$ types can display radically different kinetic properties, requiring different mathematical models. However, the construction of Hodgkin-Huxley-like (HH-like) models is generally complex and time consuming due to the number of parameters, their tuning and having to choose functional forms to model gating. In addition to the between-$K_v$ type heterogeneity, there can be significant within-$K_v$ type kinetic heterogeneity between different cells with genetically identical channels. Since HH-like models do not account for such variability, extensions to it are necessary. We use scientific machine learning (SciML), the integration of machine learning methodologies with existing scientific models, and non-linear mixed effects (NLME) modelling to bypass the limitations of HH-like modelling. NLME is a modelling methodology that takes into account both within- and between-subject variability. These tools allowed us to complement the HH-like modelling and construct a unified SciML HH-like model that fits the recordings from 20 different $K_v$ types. The unified SciML HH-like model produced closer fits to the data compared to a set of seven previous HH-like models and was able to represent the highly heterogeneous data from different cells. Our model may be the first step in producing a SciML foundation model for ion channels that would be capable of modelling the gating kinetics of any ion channel type.

## Author summary

Ion channels are complex molecules embedded in the membranes of neurons – the cells responsible for signal propagation and processing in the brain. Ion

**Data availability statement:** All of the data is freely publicly available at https://channelpedia.epfl.ch/. The specific raw data used in this study is available at https://doi.org/10.7488/ds/8052 and the code is available at https://github.com/dom-linkevicius/SciMLHHModels.jl.

**Funding:** D.L. was funded by a PhD stipend by the United Kingdom Research and Innovation (grant EP/S02431X/1), UKRI Centre for Doctoral Training in Biomedical AI at the University of Edinburgh, School of Informatics. The funders had no role in study design, data collection and analysis, decision to publish, or preparation of the manuscript. D.C.S.,M.I.S. and A.C. did not receive any specific funding for this work.

**Competing interests:** The authors have declared that no competing interests exist.

channels can open and close in response to various types of stimuli, in particular the voltage difference across the cell membrane. Computational modelling, usage of mathematical techniques to represent a system and algorithmically solve for its dynamics, has been previously used to understand the dynamics of voltage-gated ion channels. However, computational modelling of voltage-gated ion channels requires costly and complex optimization routines to fit their structure and parameters. We utilize two tools new to the modelling of voltage-gated ion channels – scientific machine learning and non-linear mixed effects modelling – to bypass some limitations associated with the existing methods.

By using scientific machine learning and non-linear mixed effects modelling we were able to create a unified model capable of modelling the gating dynamics of 20 different ion channels. This is in stark contrast to the existing modelling approaches, where each channel requires its own model. Moreover, our unified model performed better than seven existing ion channel gating models. Therefore, the tools we used and the model we created is a significant step forward in facilitating the modelling of ion channel gating. Future work could include even more ion channels types within the scope of our unified model.

## Introduction

Ion channels are ubiquitous in cellular membranes and essential for various forms of signal transduction. There are multiple different mechanisms by which ion channels can be activated, such as different ligands, mechanical stimulation, and voltage, whereas some channels are passively open [1]. Voltage-gated ion channels are important because they are one of the main means of signal conduction in the nervous system and neural cells [2,3]. An important feature of a large number of voltage-gated ion channels (with some exceptions, e.g., the non-specific HCN channel [4]) is their permeability to one dominant ion type, with sodium, potassium and calcium being the most prominent [1]. Moreover, even among voltage-gated channels that are permeable to the same ion, there is significant functional and structural heterogeneity [5]. For example, there are 40 different voltage-gated potassium channels expressed in the mammalian brain, divided into 12 subfamilies called $K_v$1-12 [6]. Different voltage-gated ion channel types show different voltage dependencies, making their role in electrical signalling functionally distinct, yet overlapping [7].

One way to understand the functional complexity resulting from the abundance of different voltage-gated ion channel types is computational modelling. Mathematical models of ion channels found in the central nervous system can be used in compartmental models of neurons to elucidate the importance and interactions between various channel types, as well as the interaction between electrical and biochemical signalling [8,9]. One of the most prominent and widespread paradigms of voltage-gated ion channel modelling was first published in the pioneering work of Hodgkin and Huxley [2]. Hodgkin and Huxley used various experimental

manipulations to characterize the voltage dependencies of the sodium and potassium channel gating in the squid giant axon. The essence of their method is computational modelling of hypothesised gating variables via differential equations. Adapting the notation in [10], two voltage and temperature-dependent functions describe the dynamics of the $i$th gating variable $m_i$: the steady-state probability of the gate being open $m_{i,\infty}(V, T)$ and the time-constant of the kinetics $\tau_i(V, T)$, where $V$ is the membrane potential and $T$ is the temperature. These functions are used in an ordinary differential equation system which can be solved numerically to obtain the dynamics of the gating variables over time $m_i(t)$. The gating dynamics can then be used to obtain the current conducted over time $I(t)$. There are many possible functional forms used for $m_{i,\infty}(V, T)$ and $\tau_i(V, T)$ (see various models in https://modeldb.science/ [8] for examples).

Even though the HH modelling paradigm was a breakthrough, aspects of its application can be complex: 1) setting up an appropriate $m_i$ structure often requires either discrete optimization routines or manual hand-tuning, including piecewise definitions (e.g., see $K_v3.4$ in https://modeldb.science/229585); 2) the assumption of independence between the gating variables that may make it difficult to fit the data. The first difficulty is not unique to the HH paradigm, it also applies to Markov models, which are a generalization of the kinetic schemes used by Hodgkin and Huxley [2]. Markov models offer more flexibility, but can often result in parameter and kinetic scheme structural unidentifiablity [11]. The difficulties of the Hodgkin-Huxley model were highlighted with the publication of a data set described in Ranjan et al. [5].

The data recorded by Ranjan et al. [5] (https://channelpedia.epfl.ch/expdata) contains automated voltage-clamp recordings of 40 different $K_v$ channels, one at a time expressed in CHO cells. They used a set of voltage-clamping protocols to probe the activation, inactivation, deactivation and recovery of $K_v$ channels, as well as their response to ramp and action potential (AP)-like stimuli. One of the key insights from the Ranjan et al. [5] data is the inherent kinetic heterogeneity of some $K_v$ subtypes: even though the $K_v$ channels expressed in the cells were genetically the same, the recordings showed significant variability for some channel types, e.g., for $K_v1.3$ or $K_v3.4$. The data showed variability in what using the HH paradigm would be modelled as $m_{i,\infty}$, as well as $\tau_i$.

The challenge posed by the data in Ranjan et al. [5] to the HH and Markov paradigms is that the basic models have one set of parameters and are not built for handling heterogeneity, i.e., different functional forms for $m_{i,\infty}(V, T)$ and $\tau_i(V, T)$ and number of gates $m_i$ with different functions may be required for different cells. Fitting a model to a single current recording is a challenging task; fitting many such models via the classical HH paradigm may be prohibitively expensive. Although there are purpose-built approaches to identifying the Markov gating structure of a given current recording [12], to the best of our knowledge, no approach currently used in computational neuroscience can efficiently handle the challenge of data heterogeneity.

To address the challenge of heterogeneous channel behaviour, we apply modelling tools used in pharmacokinetics and pharmacodynamics (PK/PD), where heterogeneous populations of patients and complex dynamics are commonplace. The nonlinear mixed effects (NLME) modelling [13] framework is common in PK/PD literature and is well-suited to model complex dynamics in heterogeneous populations. NLME is a hierarchical framework with the population level parameters (fixed effects) at the highest level and individualized parameters (random effects) at the lowest level (see Fig 1). This structure allows to both individualize predictions in the presence of significant individual differences (for example, people in clinical trials), as well as extract average, population level models useful for future predictions. Moreover, the NLME approach allows for arbitrary nonlinear functions to be used, so long as the gradient vectors with respect to both fixed and random effects can be calculated. NLME modelling has been applied widely in PK/PD (e.g., see [14]), and infectious disease modelling [15], but it is also gaining popularity in other domains, for example, individualized predictions of mood [16], repeated measurements of individualized gaze estimation [17] and many other applications where the hierarchical structure is more faithful to reality than a single level structure.

The feature of the mixed effect frameworks to include flexible functions lends itself to integration with machine learning approaches. One of the first applications of mixed effects along with neural networks used a ResNet architecture, with the resulting approach called a mixed effects network (MeNet) [17]. Using a MeNet authors processed gaze estimation image data and then used the extracted features within a linear mixed effects framework. Since the gaze estimation data they

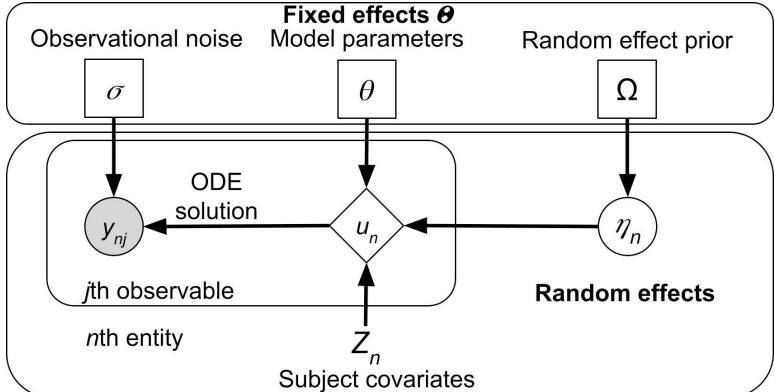

**Fig 1. Visual representation of an NLME model, rectangle nodes in the top box denote parameters (fixed effects), circles denote random quantities which are either latent (unfilled) or observed (filled), diamonds are deterministic given the inputs, and nodes without a border are constant.**

used were obtained by repeated measurements from the same participants, taking into account this dependency between the data resulted in up to 20% improvements in errors over the state of the art. A generalized version of the MeNet approach is neural mixed effects (NME) [16]. In NME, instead of placing a neural network as a feature extractor (a fixed effect), there are random effects on the neural network weights, using both person-generic parameters $\bar{\theta}$ and person-specific parameters $\theta_n$, using $\bar{\theta} + \theta_n$ as the actual neural network weights. NMEs outperformed MeNets on six different tasks. Therefore, the NME approach, individualization of the neural network weights, provided additional benefits to performance. Finally, the DeepNLME approach [18], instead of putting a random effect on each parameter of the neural net, uses random effects as inputs to the neural network, in addition to its regular inputs. Thus the neural net weights are fixed effects, yet the resulting functions are individualized for the $n$th observed entity. Approaches that integrate other machine learning models, such as random forests, can also be used combination with mixed effects [19–22].

Given nonlinear functions powerful enough to model any $m_{i,\infty}(V, T)$ and $\tau_i(V, T)$ present in the data and enough gates $m_i$, the NLME approach would serve as a suitable tool to handle the heterogeneity observed in the Ranjan et al. [5] data. We use NLME and neural networks, which are universal function approximators [23], to represent $m_{i,\infty}(V, T)$ and $\tau_i(V, T)$ and fit a single HH-like model for all K$_v$ channels for which Ranjan et al. [5] provide data. We first fit a unified K$_v$ channel model by pooling data from different channel types into a single data set, relying on the power of neural networks to provide functional flexibility and the random effects to capture the heterogeneity present in the data. We then utilize the channel type and recording temperature information of individual recordings to predict a portion of variability inherent to the channel type and temperature that was initially captured by the random effects. This step also allows the derivation of average models for different K$_v$ types. We then compare our unified model performance against multiple existing single channel model baselines. We show that the predictions of our unified model result in lower RMSE values than ones from models used in existing publications, even if random effects are added to their parameters. This comparison proves that our unified model that uses neural networks to represent $m_{i,\infty}(V, T)$ and $\tau_i(V, T)$ learns representations that can model the K$_v$ dynamics in response to protocols presented in the data more accurately than the bespoke models. Moreover, the benefits of having a unified model are shown via the superior performance of the unified model against multiple separate models using the same neural network-based approach. We hypothesize that the unified approach outperforms the multiple single channel modelling approach due to the sharing of information between similar K$_v$ types, especially when data is more limited, and finding more universal $m_{i,\infty}(V, T)$ and $\tau_i(V, T)$ forms. Finally, we analyse the $m_{i,\infty}(V, T)$ and $\tau_i(V, T)$ functions, and the temperature dependence of different K$_v$ types.

Our results are practically useful in many ways, for example, by providing a significantly simpler and more efficient way to represent different $K_v$ models, by providing a workflow of voltage-gated ion channel model development that is significantly more powerful than the previously used methods and a set of heterogeneous $K_v$ channel fits that are usable in traditional computational neuroscience simulators, such as NEURON [24]. Finally, our results point to a set of new higher level possibilities, such as a unified channel model representing both potassium and sodium channels, as well as being able to more easily integrate information of how different cellular factors affect channel dynamics.

## Methods

### Raw data

We downloaded the data for all cells that were classified as active by Ranjan et al. [5] for further processing locally. For a full description of the data and the recording conditions see the original publication (see S1 Table for the cell IDs used). The raw current recording data used in this study has been downloaded in the neurodata without borders (`.nwb`) format from https://channelpedia.epfl.ch/expdata for the following 20 channel types: $K_v1.1$, $K_v1.2$, $K_v1.3$, $K_v1.4$, $K_v1.5$, $K_v1.6$, $K_v1.8$, $K_v2.1$, $K_v2.2$, $K_v3.1$, $K_v3.1$, $K_v3.3$, $K_v3.4$, $K_v4.1$, $K_v4.2$, $K_v4.3$, $K_v10.1$, $K_v10.2$, $K_v12.1$, and $K_v12.3$. These channel types were chosen based on a preliminary visual inspection of the quality of recordings (e.g., absence of systematic artefacts) and the existence of multiple recordings at each of the different temperatures (15°C, 25°C and 35°C, the only temperatures used in Ranjan et al. [5]). For reproducibility, the subset of the Ranjan et al. [5] data we used is stored on the University of Edinburgh DataShare platform (https://doi.org/10.7488/ds/8052).

**Voltage-clamp protocols used in Ranjan et al.** The current recordings in Ranjan et al. [5] were obtained by applying (often multiple times) six voltage-clamp protocols. Each protocol began with a 100ms baseline period, where in the first 40ms each cell was held at –80mV, then the holding voltage was set to –90mV for 10ms and from 50ms to 100ms the voltage was set back to –80mV. After this initial baseline period, each cell was exposed to one of six voltage clamp protocols – activation, deactivation, inactivation, recovery, ramp and AP-like stimulation – meant to probe different $K_v$ channel properties. Each protocol consisted of a specific number of sweeps where a single feature of the protocol was varied. After each sweep, the cell was held at –80mV for the final 100ms. Generally, after the initial activation protocol, the protocols in Ranjan et al. [5] were applied in a consistent order (potentially multiple times): activation, ramp, deactivation, action potential (AP), inactivation. After the repetition of these five protocols, the recovery protocol was applied, potentially multiple times. The `Pumas.jl` package used to fit the model to the data (see below) does not currently support continuously applied time-varying negative stimuli, such as the AP and ramp protocols. Therefore, we only used four of the six protocols – activation, deactivation, inactivation and recovery, described briefly below; for full description, see Ranjan et al. [5].

**Activation.** The activation protocol consists of 18 sweeps at different holding voltages meant to measure the steady state activation levels of $K_v$ channels. Specifically, after the baseline period, cells were held between –90mV and +80mV (10mV increments) for 500ms.

**Deactivation.** The deactivation protocol consists of 12 sweeps meant to probe the deactivation kinetics of $K_v$ channels. Each sweep includes two different voltage levels: during the first 300ms the cells were clamped at +70mV, whereas for the subsequent 200ms the cells were held at voltages between –80mV to +30mV (10mV increments).

**Inactivation.** The inactivation protocol consists of 12 sweeps meant to probe the inactivation kinetics of $K_v$ channels. Each sweep includes two different voltage levels. During the first 1500ms the cells were held at voltages between –40mV to +70mV (10mV increments). Afterwards, the holding voltage is switched to +30mV for 100ms.

**Recovery.** The recovery protocol consists of 16 sweeps intended to investigate the kinetics of $K_v$ recovery after activation. For the first 1500ms the cells were held at +50mV to induce channel activation and, potentially, inactivation. Then, each cell was held at –80mV for a time period ranging between 50ms to 2300ms (150ms increments) during which

the channels recovered. Finally, the cells were held at +50mV for 200ms to measure the levels of channel recovery after the recovery period.

## Data processing

In order to make the data in Ranjan et al. [5] suitable for model training we undertook a series of data processing steps (Fig 2). We describe each step in more detail below. Briefly, we filtered out inconsistent data, then applied a time series smoothing algorithm to reduce noise levels, followed by setting the current baseline to 0, normalization of the current, rescaling, exclusion of systematic data artefacts and downsampling. The hyperparameters of the data processing steps were chosen such that the processed data would retain as much visual similarity to the original data as possible. All data processing, modelling and analyses in this paper were performed using the `Julia v1.10.4` programming language [25] due to the superior ordinary differential equation solver speeds of `DifferentialEquations.jl` compared to packages available in other programming languages, like Python or MATLAB [26,27], as well as the existence of a performant nonlinear mixed effects package with machine learning capabilities – `Pumas.jl` [28] and `DeepPumas.jl` [29]. The code used to preprocess the data is available at https://github.com/dom-linkevicius/SciMLHHModels.jl.

**Filtering out inconsistent data.** We included only the second repetition of the activation protocol and the first repetition of each other protocol for further processing instead of averaging the different repetitions. We chose this approach due to frequent inconsistencies of current amplitudes or time constants between subsequent repetitions. We assumed that the earlier repetitions are of higher quality due to suffering from rundown less.

First, we visually inspected the second repetition of the activation protocol for all of the cells and if any systematic irregularities were detected, we manually excluded the cells from further usage. This left a total of 2969 cells for further

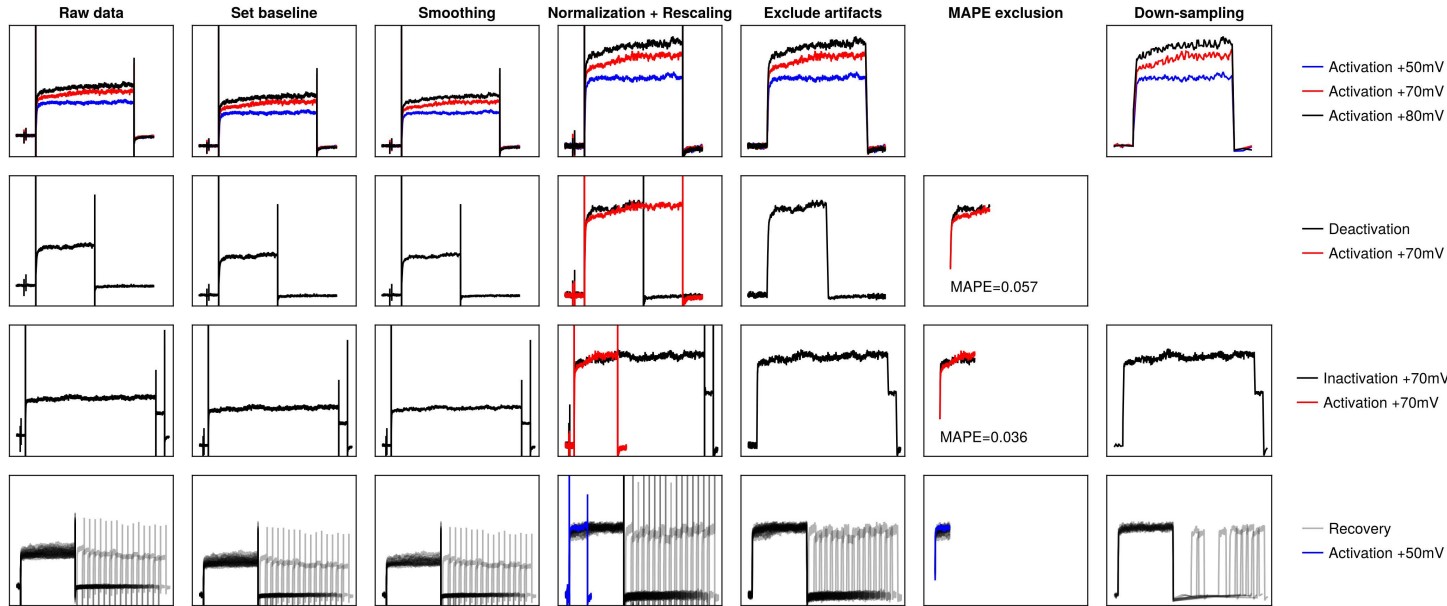

**Fig 2. Visual representation of data processing pipeline.** Each row represents the processing undertaken by some sample sweeps for each protocol: top row – activation, second row – deactivation, third row – inactivation, fourth row – recovery. The legends for each row are given in the last column. The plot for MAPE exclusion for the activation protocol is not shown because this step is not applied to the activation protocol. The down-sampling was not applied to the deactivation protocol because MAPE > 0.05. The normalization and rescaling column contains the relevant activation trace to which the other protocols were rescaled to.

processing. For a more detailed summary of the number cells or sweeps left after the data processing see Tables 1 and 2 respectively.

Since the activation sweeps were recorded first, we treated them as the standard reference point against which subsequent data is to be compared and scaled to. We compared the sweeps from other protocols to the relevant parts of the activation protocol using the mean absolute percentage error (MAPE), which is a well-known comparison measure in time series forecasting [30,31]. Here $M$ is the length of the time series being compared, $x$ is the reference time series and $y$ is the time series being evaluated:

$$\text{MAPE}(x, y) = \frac{1}{M} \sum_{i=1}^{M} \left| \frac{x_i - y_i}{x_i} \right|$$

(1)

Using cell and sweep-specific references we then excluded any subsequent sweeps that were not consistent with the relevant activation sweep according to the following criteria:

- **Deactivation** – after the initial baseline period of 100ms, for the next 300ms in all sweeps in the deactivation protocol the cells are held at +70mV. If after setting the baseline to 0 (by subtracting the mean of the current during the first 40ms of the baseline period), and scaling of the peak current to match the activation protocol peak at +70mV, MAPE > 0.05, the sweep is not used.

**Table 1. Description of the initial and the processed number of cells per channel type.**

| Channel type | Initial number of cells | | | | Number of cells after preprocessing | | | |
|---|---|---|---|---|---|---|---|---|
| | Total | 15°C | 25°C | 35°C | Total | 15°C | 25°C | 35°C |
| All types | 3171 | 807 | 1614 | 750 | 662 | 171 | 330 | 161 |
| K$_v$1.1 | 298 | 68 | 163 | 67 | 55 | 14 | 27 | 14 |
| K$_v$1.2 | 210 | 57 | 110 | 43 | 45 | 12 | 23 | 10 |
| K$_v$1.3 | 363 | 73 | 203 | 87 | 74 | 15 | 41 | 18 |
| K$_v$1.4 | 229 | 57 | 120 | 52 | 47 | 12 | 24 | 11 |
| K$_v$1.5 | 249 | 37 | 179 | 33 | 52 | 8 | 36 | 8 |
| K$_v$1.6 | 134 | 35 | 62 | 37 | 28 | 7 | 13 | 8 |
| K$_v$1.8 | 91 | 24 | 49 | 18 | 20 | 6 | 11 | 3 |
| K$_v$2.1 | 102 | 32 | 56 | 14 | 23 | 7 | 12 | 4 |
| K$_v$2.2 | 115 | 20 | 63 | 32 | 25 | 4 | 14 | 7 |
| K$_v$3.1 | 120 | 42 | 45 | 33 | 26 | 9 | 9 | 8 |
| K$_v$3.2 | 153 | 34 | 90 | 29 | 33 | 8 | 18 | 7 |
| K$_v$3.3 | 233 | 36 | 123 | 74 | 49 | 8 | 26 | 15 |
| K$_v$3.4 | 154 | 41 | 72 | 41 | 33 | 9 | 15 | 9 |
| K$_v$4.1 | 102 | 31 | 49 | 22 | 21 | 6 | 11 | 4 |
| K$_v$4.2 | 96 | 32 | 33 | 31 | 19 | 7 | 7 | 5 |
| K$_v$4.3 | 120 | 39 | 63 | 18 | 24 | 8 | 13 | 3 |
| K$_v$10.1 | 113 | 38 | 49 | 26 | 25 | 8 | 11 | 6 |
| K$_v$10.2 | 61 | 19 | 13 | 29 | 15 | 4 | 4 | 7 |
| K$_v$12.1 | 159 | 60 | 50 | 49 | 34 | 12 | 11 | 11 |
| K$_v$12.3 | 69 | 32 | 22 | 15 | 14 | 7 | 4 | 3 |

**Table 2. Number of sweeps of the data remaining after filtering of inconsistent data and irregularities.**

| Channel type | Total # of sweeps | 15°C | 25°C | 35°C | Activation | Deactivation | Inactivation | Recovery |
|---|---|---|---|---|---|---|---|---|
| All types | 87640 | 25788 | 45535 | 16317 | 52020 | 17867 | 10785 | 6968 |
| K$_v$1.1 | 7024 | 2168 | 3417 | 1439 | 4716 | 1378 | 618 | 312 |
| K$_v$1.2 | 6010 | 1848 | 3165 | 997 | 3600 | 1462 | 503 | 445 |
| K$_v$1.3 | 8323 | 2042 | 4628 | 1653 | 6120 | 1403 | 493 | 307 |
| K$_v$1.4 | 5040 | 1169 | 2806 | 1065 | 3942 | 936 | 145 | 17 |
| K$_v$1.5 | 8788 | 1696 | 6234 | 858 | 4356 | 2030 | 1090 | 1312 |
| K$_v$1.6 | 3837 | 1312 | 1703 | 822 | 2394 | 817 | 376 | 250 |
| K$_v$1.8 | 1116 | 514 | 380 | 222 | 972 | 46 | 91 | 7 |
| K$_v$2.1 | 4552 | 1578 | 2471 | 503 | 1800 | 1180 | 699 | 873 |
| K$_v$2.2 | 4071 | 736 | 2464 | 871 | 2034 | 1134 | 475 | 428 |
| K$_v$3.1 | 4943 | 1882 | 1929 | 1132 | 2034 | 1286 | 949 | 674 |
| K$_v$3.2 | 6663 | 1678 | 4045 | 940 | 2700 | 1650 | 1259 | 1054 |
| K$_v$3.3 | 8087 | 1626 | 4595 | 1866 | 4104 | 1993 | 1180 | 810 |
| K$_v$3.4 | 3421 | 1275 | 1308 | 838 | 2430 | 527 | 364 | 100 |
| K$_v$4.1 | 2028 | 922 | 870 | 236 | 1512 | 319 | 141 | 56 |
| K$_v$4.2 | 1318 | 697 | 381 | 240 | 1134 | 53 | 117 | 14 |
| K$_v$4.3 | 2557 | 1009 | 1322 | 226 | 1908 | 477 | 128 | 44 |
| K$_v$10.1 | 3165 | 991 | 1547 | 627 | 1926 | 596 | 635 | 8 |
| K$_v$10.2 | 1238 | 386 | 242 | 610 | 1026 | 2 | 204 | 6 |
| K$_v$12.1 | 4161 | 1627 | 1638 | 896 | 2286 | 563 | 1061 | 251 |
| K$_v$12.3 | 1298 | 632 | 390 | 276 | 1026 | 15 | 257 | 0 |

- **Inactivation** – after the initial baseline period of 100ms, cells are held at voltages ranging from –40mV to +70mV, which is a subset of the voltages used in the activation sweeps. Therefore, we took the first 500ms post-baseline from the inactivation sweeps and the activation sweeps held at the same voltage. We then set both of their baselines to 0 (by subtracting the mean of the current during the first 40ms of the baseline period) and scaled both inactivation and activation sweeps by the ratio of the activation peak at +80mV divided by the peak of the respective sweep (inactivation or activation). If MAPE > 0.05 for the inactivation protocol compared against the matching activation protocol, the inactivation sweep was not used. We chose to scale both sweeps to the activation peak at +80mV, which introduces a systematic bias, instead of the peak at the matching activation voltage to avoid introducing additional noise which often resulted in inconsistencies in inactivation peaks, where clamping to lower voltages resulting in higher current peaks compared to higher voltages.

- **Recovery** – after the initial baseline period of 100ms, cells are held at +50mV for the first 1500ms. Therefore, we took the first 500ms of that 1500ms and compared against the 500ms of activation protocol at +50mV. We set the baseline of both sweeps to 0, scaled both sweeps by the ratio of activation peak at +80mV divided by the peak of the given sweep and calculated the MAPE values. Moreover, there were multiple cases where the recovery sweeps were internally inconsistent; some traces showed significantly different current values at the end the 1500ms. Therefore, we calculated the median current value at 1500ms post-baseline over all the sweeps of a given cell and we used recovery sweep only if MAPE < 0.05 and the current value at 1500ms post-baseline was within one standard deviation of the median.

These consistency checks were necessary to ensure that we kept only the sweeps that were internally consistent with the activation sweeps, for example there were no significant differences in time constants or large recording artifacts, etc. Usage of inconsistent data would have significantly hampered model training by requiring additional model mechanisms capable of accounting for the inconsistencies.

 

**Smoothing.** After the filtering of the sweeps inconsistent with the second repetition of the activation protocol, we smoothed the data using the Julia `KissSmoothing.jl v1.0.8` package (see https://github.com/francescoalemanno/KissSmoothing.jl). It implements a smoothing function called `denoise` which transforms the signal using the discrete cosine transform and convolves it with an Gaussian function (similar to the method outlined in [32]). We use `denoise` with default parameters, except for setting the smoothing intensity to `factor=1.0`.

**Setting the baseline.** Since the raw current recordings were often offset from 0 at −80mV, when there should be no significant potassium current (due to the voltage being close to the reversal potential and channels generally being closed), we calculated the mean of the first 40ms of each sweep and subtracted it from the full sweep to set the mean during the baseline period to 0.

**Normalization.** Since we were only interested in the channel gating dynamics, we normalized each sweep by the current peak $I_{max}$ of activation sweep when the cell was held at +80mV. This allows us to model only the gating dynamics, reducing the number of parameters that need to be fit.

**Rescaling.** For each sweep of each protocol other than activation, it was rescaled by ratio of the matching activation sweep peak current divided by the given sweep peak current before normalization by $I_{max}$. Concretely, each sweep of the deactivation protocol was rescaled by the activation sweep peak when the cell is held at +70mV. Each trace of the inactivation protocol was rescaled by the matching activation sweep. Finally, each sweep of the recovery protocol was rescaled by the activation sweep peak when the cell is held at +50mV.

**Excluding systematic recording artifacts.** A significant number of sweeps showed fast negative deflection artifacts whenever voltage-clamping level was changed. This behavior does not arise from channel gating dynamics, therefore, after taking the previously described data processing steps, we excluded any points that are more negative than −0.01. This is to avoid the model being penalized for not fitting the data it by construction can not reproduce.

**Down-sampling of sweeps.** In order to speed up model training we down-sampled all the sweeps by using the M4 down-sampler. The M4 down-sampler was chosen because of its simplicity, efficiency and preservation of extrema [33]. More specifically, for a given time-series $x$ and number of bins $b$, the M4 down-sampling algorithm starts by dividing $x$ into $b$ non-overlapping bins and then takes the first and the last values in that bin, as well as the maximal and the minimal remaining values. Therefore, the M4 down-sampler reduces the length of a time-series from the initial length $k$ to $4b$.

Each initial 100ms baseline recording was reduced from 1000 to 9 points: 3 points from the first 40ms, 3 points from 40ms to 50ms and 3 points from 50ms to 98ms, all spaced equidistantly within each of these periods. The final 100ms was also down-sampled to 3 equidistant points. For the activation protocol, the 4000 points starting from 100ms to 600ms were down-sampled to 120 points. For the deactivation protocol, the 3000 points between 100ms to 400ms were down-sampled to 72 points, while the points between 400ms to 600ms were down-sampled to 48 points. For the inactivation protocol the 15000 points between 100ms and 1500ms were down-sampled to 360 points, whereas the 1000 points between 1600ms to 1700ms were down-sampled to 24 points. Finally, for the recovery protocol, the 15000 points between 100ms and 1600ms were down-sampled to 360 points, the period between the initial +50mV and the test +50mV that was of variable length was down-sampled to 3 points spaced equidistantly, and the second +50mV period of 150ms was down-sampled to 48 points. Down-sampling was the final step of our data processing pipeline. The data remaining at this point was split into training, validation and test data sets (described below). The full raw and the processed data are available to be downloaded at [34]. Next we move on to describe the general nonlinear mixed effects modelling approach.

### Nonlinear mixed effects (NMLE) modelling

We adapt the definitions of NLME provided in [28,35]. The NLME modelling framework comprises a two level hierarchical structure (Fig 1) with fixed effects Θ at the upper level which do not vary between recorded entities. Fixed effects can be broadly grouped into

- model parameters $\theta$

- random effect prior distribution parameters $\Omega$

- observation model noise parameters $\sigma$

The lower level consists of random effects $\eta_n$ which account for the inter-individual variability of the observations by individualizing the model parameters $\theta$ for the $n$th individual. In this study the inter-individual variability is between the recorded currents $y_n$ for the $n$th cell. Furthermore, a set of covariates $Z_n$ (which are known at the outset) are associated with the $n$th cell, namely, the temperature of the recording (but see below), as well as the channel type.

These three sets of values ($\Theta$, $\eta_n$ and $Z_n$) are collated via the parameter model $g$ (note the similar but distinct notation $\bar{g}$ for the maximal channel conductance) into the dynamical parameter vector $p_n$ of the $n$th cell

$$p_n = g(\Theta, Z_n, \eta_n) \tag{2}$$

The dynamical parameters $p_n$ are then fed into the structural model, e.g., an ordinary differential equation (ODE) system:

$$u'_n = f(u_n, p_n, t) \tag{3}$$

where $u_n$ are the dynamical variables being solved for, e.g., the gating variables in the Hodgkin-Huxley formalism, and where $u'$ denotes the time derivative of $u$. The final step is to link the numerical solution of the ODE system to the experimentally observed quantities $y_{nj}$, where $j$ denotes the number of different observed quantities for the $n$th individual. In this study the only observable quantity is the recorded normalized current defined in Equation 15. After numerically solving Equation 3, the observable quantities are derived from the values obtained numerically and passed through a Gaussian observational model to account for observational noise, giving $y_{nj}$ (in this study there is only one observed quantity, therefore $j$ is omitted)

$$p(y_n(t) \mid \Theta, \eta_n) = \mathcal{N}\left(\frac{I_n(t)}{I_{n_{\max}}}, \sigma\right) \tag{4}$$

where $\frac{I_n(t)}{I_{n_{\max}}}$ is defined in Equation 15. In all following equations the dependence of $y_n$ on time will be omitted to reduce visual clutter.

There are many ways to fit NLME models, both frequentist and Bayesian [36]. In this study we used both the maximum a posteriori (MAP) conditional log-likelihood objective which can be stated as

$$\Theta^*, \eta^* = \arg\max_{\Theta, \eta} \left( p(\Theta) \cdot \prod_{i=1}^{N} p(y_n \mid \Theta, \eta_n) \cdot p(\eta_n \mid \Theta) \right) \tag{5}$$

as well as the maximum a posteriori (MAP) first order conditional estimate (FOCE) of the marginal likelihood [10]

$$\Theta^* = \arg\max_{\Theta} \left( p(\Theta) \cdot \prod_{n=1}^{N} p(y_n \mid \Theta) \right)$$
$$= \arg\max_{\Theta} \left( p(\Theta) \cdot \prod_{n=1}^{N} \int p(y_n \mid \Theta, \eta_n) \cdot p(\eta_n \mid \Theta) \, d\eta_n \right) \tag{6}$$

with the random effects for individual cells set to their empirical Bayes estimate (EBE$_n$ values defined as

$$\text{EBE}_n = \eta_n^* = \arg\max_{\eta_n} \left( p(y_n \mid \Theta = \Theta^*, \eta_n) \cdot p(\eta_n \mid \Theta = \Theta^*) \right)$$

(7)

In both definitions $\Theta^*$ is the mode of the fixed effects and $p(\Theta)$ is the fixed effect prior distribution. We specify how we used both of these objectives below.

## K$_v$ channel models

In this study we use four types of models which we evaluate and fit to the Ranjan et al. [5] data, all of which follow the general HH equations for the gating variables $m_i$ which form the function $f$ in Equation 3:

$$\frac{dm_i}{dt} = \frac{m_{i,\infty}(V, T) - m_i}{\tau_i(V, T)}$$

(8)

This can be interpreted as arising from first order reactions of $m_i$ transitioning between a closed and an open state, which, when normalised to 1 and denoting $m_i$ as the open state

$$\frac{dm_i}{dt} = \alpha_i(V, T)(1 - m_i) - \beta_i(V, T)m_i$$

(9)

where $\alpha_i(V, T)$ and $\beta_i(V, T)$ are voltage and temperature dependent reaction rates for the $i$th gating particle. Then $m_{i,\infty}(V, T)$ and $\tau_i(V, T)$ are defined as

$$m_{i,\infty}(V, T) = \frac{\alpha_i(V, T)}{\alpha_i(V, T) + \beta_i(V, T)}$$

(10)

$$\tau_i(V, T) = \frac{1}{\alpha_i(V, T) + \beta_i(V, T)}$$

(11)

noting that generally $m_{i,\infty}(V, T) \in [0, 1]$ and $\tau_i(V, T) \in (0, \infty)$. Furthermore, given two functions $m_{i,\infty}(V, T)$ and $\tau_i(V, T)$ that abide by the aforementioned restrictions, it is possible to derive $\alpha_i(V, T)$ and $\beta_i(V, T)$

$$\alpha_i(V, T) = \frac{m_{i,\infty}(V, T)}{\tau_i(V, T)}$$

(12)

$$\beta_i(V, T) = \frac{1 - m_{i,\infty}(V, T)}{\tau_i(V, T)}$$

(13)

Equation (8) can be solved numerically to obtain the current $I(t)$ conducted by the channels over time. $I(t)$ is the product of the gating variables $m_i(t)$, the maximum channel conductance $\bar{g}$, the ionic reversal potential $E$ and the gating variable power $n_i$

$$I(t) = \bar{g}(V(t) - E) \prod_i m_i^{n_i}(t)$$

(14)

Since we are only interested in the channel gating dynamics captured by $m_i$, instead we derive the normalized current $\frac{I(t)}{I_{max}}$

$$\frac{I(t)}{I_{max}} = \frac{\bar{g}(V(t) - E)\prod_i m_i^{n_i}(t)}{\bar{g}(V_{max} - E)\prod_i m_{i,max}^{n_i}} = \frac{1}{\tilde{V}}(V(t) - E)\prod_i m_i^{n_i}(t)$$

(15)

where $\tilde{V} = V_{max} - E$, $V_{max} = 80\text{mV}$ and $\prod_i m_{i,max}^{n_i} = 1$ by definition. We next describe the four types of HH models that we used, which differ in their scope, the functional forms used for $m_{i,\infty}(V, T)$ and $\tau_i(V, T)$, the inclusion of the gating power $n_i$ and the inclusion of random effects.

The first model type, which we call the classical HH models, is a set of $K_v$ models from https://modeldb.science/. Since a systematic evaluation of existing $K_v$ channel models is outside the scope of this work, we selected a number of models based on a preliminary test – whether the classical model errors in fitting the Ranjan et al. [5] data for the $K_v$ channel being modelled were within an order of magnitude of the SciML models described below. Multiple models did not satisfy this criterion, but we found reasonable baseline models for the following $K_v$ channels – $K_v1.1$ [5], $K_v1.2$ [37], $K_v1.5$ [38], $K_v3.1$ [39], $K_v3.3$, $K_v3.4$ [40] and $K_v4.3$ [41] (see S2 Text for links to files used and data sources used in tuning the models). All baseline models have both activation and inactivation, except for $K_v1.5$, which has two inactivation particles, along with an activating one. All models included adjustments of functions due to temperature. The classical HH models serve as the exemplars of the class of models that are currently among the most prominent in computational neuroscience.

Since none of the classical HH models included random effects, in addition to the original models, we also analysed the same models with random effects added on their parameters. We used a standard multivariate Gaussian random effect prior and only fitted the $EBE_n$ for these models using the FOCE objective (see Equation 6). Addition of random effects allowed for a fairer comparison with the next two types of models and the baseline models, helping to isolate the impact of using alternative functional forms for $m_{i,\infty}(V, T)$ and $\tau_i(V, T)$. We call this group of models the classical HH models with random effects.

Our third model type is the individual SciML HH models. For this model type, we use neural network-based $m_{i,\infty}(V, T)$ and $\tau_i(V, T)$ functions. The models follow the classical HH formalism from Equation 8, but in this case

$$m_{i,\infty}^{(n)}(V, T_n) = \text{NN}_{m_{i,\infty}}(V, \eta_{i,m}^{(n)}, \theta_{i,\infty}, T_n)$$

(16)

$$\tau_i^{(n)}(V, T_n) = \text{NN}_{\tau_i}(V, \eta_{i,\tau}^{(n)}, \theta_{i,\tau}, T_n)$$

(17)

where $V$ is voltage, $\eta_{i,f}$ are the random effects for $f \in \{m, \tau\}$, $\theta_{i,\infty}$ and $\theta_{i,\tau}$ are neural network weights and biases of the $i$th respective neural network, $\text{NN}_{m_{i,\infty}}$ and $\text{NN}_{\tau_i}$ are specific types of neural network architecture and $T_n$ is the normalized temperature for the $n$th cell. Specifically, $\text{NN}_{m_{i,\infty}}$ is capped off with a sigmoid nonlinearity and outputs in the range of $(0, 1)$, whereas $\text{NN}_{\tau_i}$ is capped off with a softplus nonlinearity and outputs in the range of $(0, \infty)$. For all models we used $i = 2$ gates, therefore there were four vectors of neural network weights per model for a $K_v$ type, where each neural network had two hidden layers of 5 units each with L1 regularization of the weights with $\lambda = 10^{-4}$, which is equivalent to a $p(\Theta) = \text{Laplace}(0, \frac{1}{\lambda})$ prior on the fixed effects. Moreover, we used 8 random effects, two for each function.

The fourth type of model we use is the unified SciML HH model. This model type is used to model all of the $K_v$ types via the same set of fixed effects $\Theta$, largely following the neural network-based $m_{i,\infty}^{(n)}(V, T)$ and $\tau_i^{(n)}(V, T)$ functions used in the individual SciML HH models (including the same hyper-parameters). The same architecture was used in order to have the most parsimonious model that is only as large as necessary to fit the data of the different $K_v$ types. Furthermore, in this case the random effects $\eta_{i,f}$ contain two components, the truly random effects $\eta^*$ and the augmentation random

effects $\eta^\dagger$ which are added element-wise, i.e., $\eta_{i,f}^{(n)} = \eta_n^* + \eta_n^\dagger$. The augmentation random effects for the $n$th cell are based on the channel type $c_n$, as well as the normalized temperature $\bar{T}_n$. The channel type $c_n$ is encoded as a 19 dimensional one-hot vector, where all zeros would correspond to $K_v1.1$. A 19 dimensional, rather than 20 dimensional one-hot vector was chosen to slightly reduce the number of parameters in the neural network, as the expressivity of the one-hot encodings is identical. It was assumed that the normalized temperature $\bar{T}_n$ is a noisy estimate of the true temperature by adding a random effect $\eta_{temp} \sim \mathcal{N}(0, 1)$. The observation that the temperature is a noisy estimate is based on the recordings of the electrode tip temperature available in the original data set. We then pass the channel type vector and the normalized temperature to an augmentation neural network $\eta_n^\dagger = NN_{aug}(c_n, \bar{T}_n, \theta_{aug})$ where $\theta_{aug}$ is the set of neural network weights. This model structure allows the inherent heterogeneity present in the data to be accounted for, while capturing the portion of its variance attributable to different $K_v$ types and temperatures via $NN_{aug}$.

### Data splitting into training, validation and test data sets

For the individual SciML HH model training we split the processed data using a 60%/20%/20% fraction for training, validation and test. We took the appropriate fractions from each temperature for each channel type, rather than pooling the temperatures together. The validation and test data derived in this fashion were used for the validation and testing of all model types. However, the training data set for the unified SciML HH model, due to its size, required a different approach.

For the unified SciML HH model, we took seven cells for each of the three measured temperatures for each channel type, except for $K_v1.8$, $K_v4.2$, $K_v12.3$. We took seven cells because that was the maximal number of cells that we could take for each channel type such that the resulting data set would be balanced for each temperature and each channel type. This resulted in 357 cells in the training data. We left $K_v1.8$, $K_v4.2$, $K_v12.3$ out of the training data in order to establish whether the unified SciML model could generalize to channel types which were not in the training data set.

### Model fitting

We use the `Pumas.jl` [28] and `DeepPumas.jl` [29] Julia packages to solve Equations 5 and 6. `Pumas.jl` contains efficient and powerful algorithms for NLME modelling with primary focus on pharmacokinetics and pharmacodynamics, whereas `DeepPumas.jl` contains the code infrastructure necessary to incorporate neural networks into the NLME modelling. More specifically, it implements algorithms to solve equations 6, 5 and 7 and uses forward-model automatic differentiation to allow gradient-based optimization techniques. We used the L-BFGS optimization algorithm from `Optim.jl` with the gradient calculations handled by `Pumas.jl` and `DeepPumas.jl`. The ODEs defined above were solved numerically using the `DifferentialEquations.jl` package, using the `AutoVern7(Rodas5P(autodiff=true)` autoswitching solver that switches between a solver that is better suited for stiff systems, and a solver that is better suited for non-stiff systems.

Both the classical HH models with random effects and the individual SciML HH type models were fit to the individual $K_v$ channel type data, whereas the unified SciML HH model was fit to a unified data set described earlier. For the classical HH models with random effects, we fixed the $\theta$ and the $\Omega$ values and only fit the $\sigma$ and the $\eta_n$ values to data.

All fitting was done on the JuliaHub (https://juliahub.com/) cloud computing platform using nodes with 8 vCPUs and 64GB of memory. Fits took between 3 and 24 hours, depending on the $K_v$ type and the amount of data which was used. All the code that was used to define the models, run the simulations and perform the analysis is accessible at https://github.com/dom-linkevicius/SciMLHHModels.jl.git.

The first stage of fitting was shared between the individual and the unified SciML HH models. In the first step of the fitting we used the conditional likelihood objective (Equation 5) to produce an initial fit of all of the fixed effects, except for the parameters of the random effect prior $\Omega$ which was initialized to be standard normal and was held constant throughout this step of the optimization.

We ran the optimization for 300 epochs, saving the results every 15 epochs, evaluating the model performance on the validation data and selecting the set of parameters that had the best performance on the validation data. The model training generally converged well before the 300 epochs (see S1 Text). Conditional likelihood is much more numerically efficient due to $\Theta$ and $\eta_n$ being optimized jointly whereas, for example, marginal likelihood generally requires a two level optimization scheme. However, conditional likelihood requires appropriate handling of $\Omega$ to avoid overly broad random effect distributions which barely penalize extreme $\eta_n$ values and effectively result in different individual models due to the learning being offloaded mostly to the random effects.

In the second stage, which was only applied to the unified SciML HH model, we held all of the fixed effect values constant, except for the random effect prior parameters $\Omega$. Then we used the FOCE objective (Equation 6) to fit the $\Omega$. We ran the optimization for 10 iterations. The objective changed very minimally during the 10 iterations (so the quality of the data fit did not change), but the values in $\Omega$ were adjusted appropriately. Since the data fitting quality does not change in this step, we opted to omit the individual SciML HH models from this step because our main focus is on the unified SciML HH model (see Results and Discussion).

This two stage optimization procedure utilizes the numerical efficiency of the conditional likelihood by first finding the neural network weights capable of representing the functions necessary to model the current conducted via the $K_v$ channels. It also safeguards against the over-fitting of the random effects by not allowing the penalty coming from the random effect prior to become negligible via $\Omega$ assuming arbitrarily large values. The second step in the optimization pipeline only optimizes $\Omega$ using an approximation to the marginal likelihood. This step is meant to calibrate the spread of the random effects in order to properly account for the inherent heterogeneity present in the data that is not due to the differences in temperature or channel types. Under ideal circumstances of large enough compute capabilities, joint fitting of all the fixed effects using only marginal likelihood would be preferable, but was not feasible in the present study.

## Results

As described in detail in the Methods, we fitted two novel types of HH models to voltage clamp data from 20 $K_v$ channels collected by Ranjan et al. [5] and compared them against a set of classical HH models with and without random effects. The first set of seven classical HH models for different $K_v$ types was taken from the literature (see Methods). The second set of models was produced by taking the models from the first set and adding random effects on their parameters. The third set of 20 models was produced by using neural networks to represent the voltage-dependent gating functions $m_{i,\infty}(V, T)$ and $\tau_i(V, T)$ and we call them the individual SciML HH models. The final model again used neural networks to represent $m_{i,\infty}(V, T)$ and $\tau_i(V, T)$, but it was fitted to a joint data set of different $K_v$ channels, we call this model the unified SciML HH model. We present the results of both the individual and the unified SciML HH models that we fit, along with the performance of the classical HH models and the classical HH models with random effects. We only show the results of the best performing models found during the optimization. For more detailed results of model selection see S1 Text.

Fig 3 shows the comparison of the different types of HH models for $K_v$1.1. We focus on this channel type as an example because Ranjan et al. [5] fitted a HH model for it to their activation protocol data, providing the fairest available comparison. Note that the comparison is not fully fair, as our models were fitted to all of the protocols, not only activation. First of all, we reproduce Fig 8A from Ranjan et al. [5] (Fig 3A-3C), showing that their model can provide adequate fits for the activation protocol for cells that are close to the median gating kinetics. However, we also look at model predictions across cells in Fig 3D, showing the normalised currents (see Methods) for 10 sweeps of the activation protocol at +10mV (black lines) along with the respective model predictions (blue – unified SciML HH, yellow – individual SciML HH, green – classical HH without random effects, pink – classical HH with random effects). Visually, there is a clear difference between the fits for the SciML HH models and the HH models, with the classical HH models performing poorly and the SciML HH models fitting the data reasonably well. The box plots (Fig 3E) of the root mean square error (RMSE) show that the SciML HH models significantly outperform the classical HH models (for statistical test results see Table 3). Specifically, the

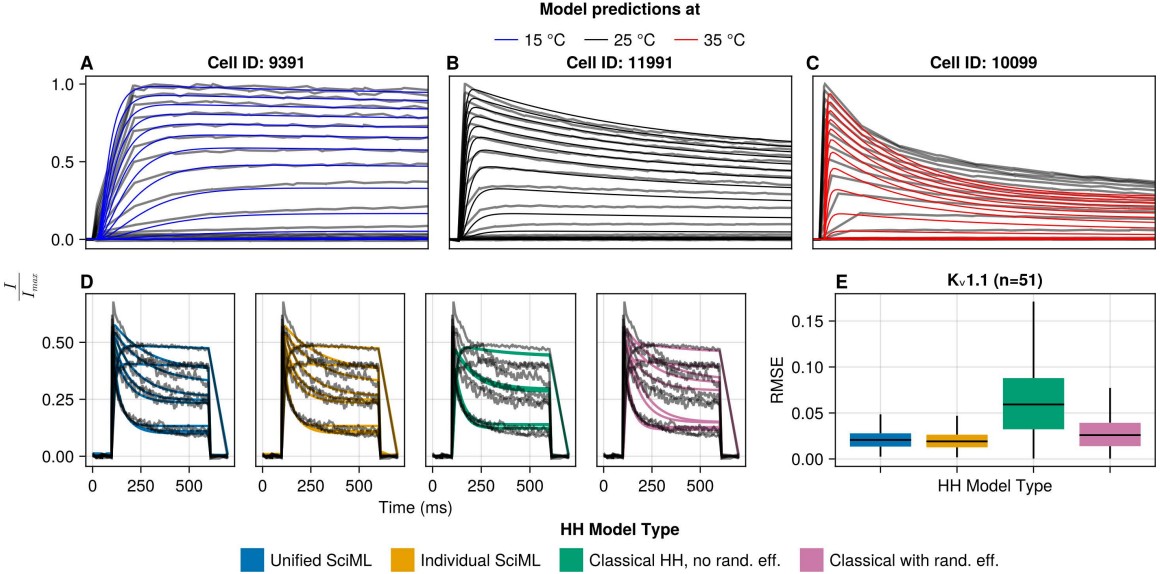

**Fig 3. Comparison of the four different HH model types for the $K_v1.1$ channel. (A–C)** Recreation of Fig 8A from Ranjan et al. [5] showing that the classical $K_v1.1$ model without the random effects can fit individual cells when their kinetics are close to the median. **(D)** Each subplot shows 10 activation sweeps at +10mV from different cells from the test data set (black) and the matching model predictions (coloured lines). The predictions made using the classical HH model without random effects (green) display variance because, even though the model parameters do not have random effects, the temperature used in it (and the $Q_{10}$ value which scales them) does. **(E)** Box plots of the test data RMSE values (at the sweep level, for example activation at +10mV, deactivation at –20mV, inactivation at –30mV).

classical HH model from Ranjan et al. [5] does not perform as well as the SciML models even when imbued with random effects, however, this could be either due to our usage of more data in model fitting or suboptimal model structure in their model. Furthermore, the individual SciML HH model significantly outperforms the unified SciML HH model, but the size of the difference is small (Table 3).

Fig 4 shows the (RMSE) box plots for different model predictions compared to normalised voltage clamp current at the protocol sweep level for individual channel types. The four types of models – unified SciML HH (blue), individual SciML HH (yellow), classical HH without random effects (green) and classical HH with random effects (pink) display a consistent pattern of performance across different $K_v$ channel types.

Firstly, the comparisons between the classical HH models with and without random effects confirms the utility of the NLME approach in modelling the $K_v$ channel data of Ranjan et al. [5]. In all seven cases ($K_v1.1$, $K_v1.2$, $K_v1.5$, $K_v3.1$, $K_v3.3$, $K_v3.4$ and $K_v4.3$) of the implemented classical HH models, the version of the model with random effects significantly outperformed the version without the random effect (see Table 3 for more details).

Secondly, comparing the individual SciML models against the classical HH models with random effects allowed us to investigate which model could predict the normalised $K_v$ current data more accurately. The functions used to model the gating variables in the classical HH models were constructed by hand, in some cases with biophysical support. We hypothesised that neural networks offer a more flexible and more powerful approach to constructing gating variable functions than the classical approach. However, note that it is not possible to disprove that the differences are caused by our usage of a richer data source than used in the construction of the classical HH models. As shown in Fig 4 and in Table 3, the individual SciML HH models fit the data significantly better than the classical HH models with random effects in all seven cases. This comparison supports the hypothesis that the flexibility of the voltage-dependent gating functions in the individual SciML HH models resulted in better fits to the Ranjan et al. data compared to a set of classical HH models.

**Table 3. Statistical comparison of performance between different models on the test data set.**

| Channel type | $\mathcal{M}_1$ | $\mathcal{M}_2$ | median($\mathcal{M}_1$-$\mathcal{M}_2$) | W statistic (n = sample size) | Cohen's d[‡] |
|---|---|---|---|---|---|
| $K_v1.1$ | uSciML | iSciML | 0.0005 | 601677* (n = 1390) | 0.102 |
| $K_v1.1$ | uSciML | cHH | -0.0331 | 38802* (n = 1390) | 1.23 |
| $K_v1.1$ | uSciML | cHH with randeff. | -0.0043 | 228727* (n = 1390) | 0.46 |
| $K_v1.1$ | iSciML | cHH | -0.0354 | 31374* (n = 1390) | 1.26 |
| $K_v1.1$ | iSciML | cHH with randeff. | -0.0048 | 160061* (n = 1390) | 0.53 |
| $K_v1.1$ | cHH | cHH with randeff. | 0.0198 | 929902* (n = 1390) | 0.92 |
| $K_v1.2$ | uSciML | iSciML | 0.0006 | 597596* (n = 1378) | 0.131 |
| $K_v1.2$ | uSciML | cHH | -0.045 | 26953* (n = 1378) | 1.42 |
| $K_v1.2$ | uSciML | cHH with randeff. | -0.0077 | 122163* (n = 1378) | 0.65 |
| $K_v1.2$ | iSciML | cHH | -0.0476 | 25419* (n = 1378) | 1.47 |
| $K_v1.2$ | iSciML | cHH with randeff. | -0.0101 | 109325* (n = 1378) | 0.76 |
| $K_v1.2$ | cHH | cHH with randeff. | 0.0319 | 911806* (n = 1378) | 1.07 |
| $K_v1.3$ | uSciML | iSciML | 0.0012 | 1098627* (n = 1857) | 0.05 |
| $K_v1.4$ | uSciML | iSciML | 0.0019 | 476142* (n = 1052) | 0.28 |
| $K_v1.5$ | uSciML | iSciML | 0.0011 | 1339387* (n = 1897) | 0.152 |
| $K_v1.5$ | uSciML | cHH | -0.0439 | 87662* (n = 1897) | 1.35 |
| $K_v1.5$ | uSciML | cHH with randeff. | -0.0038 | 379122* (n = 1897) | 0.53 |
| $K_v1.5$ | iSciML | cHH | -0.0459 | 60002* (n = 1897) | 1.4 |
| $K_v1.5$ | iSciML | cHH with randeff. | -0.0051 | 236127* (n = 1897) | 0.64 |
| $K_v1.5$ | cHH | cHH with randeff. | 0.0325 | 1688701* (n = 1897) | 1.1 |
| $K_v1.6$ | uSciML | iSciML | 0.0017 | 257410* (n = 801) | 0.36 |
| $K_v1.8$ | uSciML | iSciML | 0.0005 | 43791 (n = 410) | 0.02 |
| $K_v2.1$ | uSciML | iSciML | -0.0021 | 150324* (n = 1014) | 0.39 |
| $K_v2.2$ | uSciML | iSciML | -0.0003 | 160109 (n = 850) | 0.09 |
| $K_v3.1$ | uSciML | iSciML | -0.0003 | 229923* (n = 1094) | 0.149 |
| $K_v3.1$ | uSciML | cHH | -0.0337 | 19442* (n = 1094) | 1.07 |
| $K_v3.1$ | uSciML | cHH with randeff. | -0.0063 | 74574* (n = 1094) | 0.54 |
| $K_v3.1$ | iSciML | cHH | -0.0295 | 28610* (n = 1094) | 1.01 |
| $K_v3.1$ | iSciML | cHH with randeff. | -0.0037 | 139837* (n = 1094) | 0.43 |
| $K_v3.1$ | cHH | cHH with randeff. | 0.016 | 537257* (n = 1094) | 0.71 |
| $K_v3.2$ | uSciML | iSciML | -0.0115 | 67125* (n = 1536) | 0.76 |
| $K_v3.3$ | uSciML | iSciML | 0.0001 | 772063 (n = 1733) | 0.003 |
| $K_v3.3$ | uSciML | cHH | -0.0548 | 30199* (n = 1733) | 1.45 |
| $K_v3.3$ | uSciML | cHH with randeff. | -0.0014 | 416743* (n = 1733) | 0.25 |
| $K_v3.3$ | iSciML | cHH | -0.0546 | 19170* (n = 1733) | 1.45 |
| $K_v3.3$ | iSciML | cHH with randeff. | -0.0013 | 428191* (n = 1733) | 0.25 |
| $K_v3.3$ | cHH | cHH with randeff. | 0.0463 | 1483162* (n = 1733) | 1.32 |
| $K_v3.4$ | uSciML | iSciML | 0.0 | 123783 (n = 719) | 0.018 |
| $K_v3.4$ | uSciML | cHH | -0.0812 | 12660* (n = 719) | 0.81 |
| $K_v3.4$ | uSciML | cHH with randeff. | -0.0015 | 82089* (n = 719) | 0.03 |
| $K_v3.4$ | iSciML | cHH | -0.0787 | 9000* (n = 719) | 0.79 |
| $K_v3.4$ | iSciML | cHH with randeff. | -0.0016 | 84341* (n = 719) | 0.01 |
| $K_v3.4$ | cHH | cHH with randeff. | 0.0763 | 248774* (n = 719) | 0.79 |
| $K_v4.1$ | uSciML | iSciML | 0.0003 | 74509 (n = 510) | 0.01 |

*(Continued)*

**Table 3.** (Continued)

| Channel type | $\mathcal{M}_1$ | $\mathcal{M}_2$ | median($\mathcal{M}_1$-$\mathcal{M}_2$) | *W* statistic (*n*=sample size) | Cohen's d‡ |
|---|---|---|---|---|---|
| $K_v$4.2 | uSciML | iSciML | -0.0006 | 30289 (*n*=378) | 0.04 |
| $K_v$4.3 | uSciML | iSciML | 0.0006 | 124851* (*n*=605) | 0.045 |
| $K_v$4.3 | uSciML | cHH | -0.046 | 6232* (*n*=605) | 1.32 |
| $K_v$4.3 | uSciML | cHH with randeff. | -0.0069 | 20548* (*n*=605) | 0.65 |
| $K_v$4.3 | iSciML | cHH | -0.0459 | 4368* (*n*=605) | 1.32 |
| $K_v$4.3 | iSciML | cHH with randeff. | -0.008 | 20871* (*n*=605) | 0.64 |
| $K_v$4.3 | cHH | cHH with randeff. | 0.0346 | 175620* (*n*=605) | 1.09 |
| $K_v$10.1 | uSciML | iSciML | 0.0047 | 230959* (*n*=736) | 0.47 |
| $K_v$10.2 | uSciML | iSciML | 0.0022 | 35770* (*n*=303) | 0.2 |
| $K_v$12.1 | uSciML | iSciML | 0.0015 | 358654* (*n*=1005) | 0.14 |
| $K_v$12.3 | uSciML | iSciML | 0.003 | 33230* (*n*=307) | 0.2 |

When more than two models are evaluated for a given channel type, the highlighted row contains the two best performing models. Abbreviations for models $\mathcal{M}_1$ and $\mathcal{M}_2$: cHH – classical Hodgkin-Huxley model, cHH with randeff. – classical Hodgkin Huxley model with random effects, iSciML – individual SciML Hodgkin-Huxley model, uSciML – unified SciML Hodgkin-Huxley model. *W* - Wilcoxon rank-sum test statistic

‡ Effect sizes evaluated via Cohen's *d* follow the following rule of thumb [42]: *d* > 0.2 – small, *d* > 0.5 – medium, *d* > 0.8 – large, *d* > 1.2 – very large, *d* > 2.0 – huge. * *p* < 10⁻⁶ Bonferroni corrections were applied due to multiple comparisons.

Finally, we compare the unified SciML HH model and the individual SciML HH models. Under ideal circumstances, an individualised channel model would always outperform a more unified model due to its ability to specialize on a particular $K_v$ type. However, given that the data is limited and there are similarities between the different $K_v$ channel dynamics, a unified model may offer comparable or better performance along with the possibility of generalizing to channel types on which it was not trained. As shown in Fig 4 and in Table 3, the unified SciMLHH model outperformed the individual SciML HH models in three out of 20 cases, is outperformed in 11 out of 20 cases and in six out of 20 cases there is no significant difference. Therefore, the individual SciML HH models outperform the unified SciML HH model on the majority of $K_v$ channel types.

However, looking at the effect sizes of the differences (Table 3), the differences are generally small (Cohen's *d* > 0.2, except for $K_v$1.4, $K_v$1.6, $K_v$2.1, $K_v$3.2 and $K_v$10.1). Importantly, the unified SciML HH model performs similarly well to the individual SciML HH models on data from channel types which were not in the training data ($K_v$1.8, $K_v$4.2, $K_v$12.3, see Fig 4 subplots with light red background). Therefore, despite the statistically significant better performance of the individual SciML HH models in a higher proportion of cases, since the effect sizes are generally small and the unified SciML HH model offers a much more parsimonious representation of the data with the ability to generalize to $K_v$ channels on which it was not trained, we conclude that the unified SciML HH model is practically preferable to the individual SciML HH models. Therefore, we will restrict further analyses to the unified SciML HH model. We show some example unified SciML HH model predictions on the test data for channel types which were in the training data (Fig 5) and channel types which were not (Fig 6) to illustrate the model's ability to generalize well.

## Unified SciML Hodgkin-Huxley model diagnostics

To evaluate the model performance in an absolute sense, rather than relative to other types of models, we examine a number of diagnostics. The first set of diagnostic plots investigates the distributions of pairs of Empirical Bayes Estimates of random effects (EBEs, defined in Equation 7) obtained by fitting them to the test data. As shown in the pair plots in Fig 7 (subplots below the diagonal), the EBEs have a varying range of values, e.g., $\eta_5$ or $\eta_8$ are relatively more concentrated than $\eta_1$ or $\eta_2$. However, visually there seem to be relatively few, if any, correlations between the different $\eta_i$. The subplots

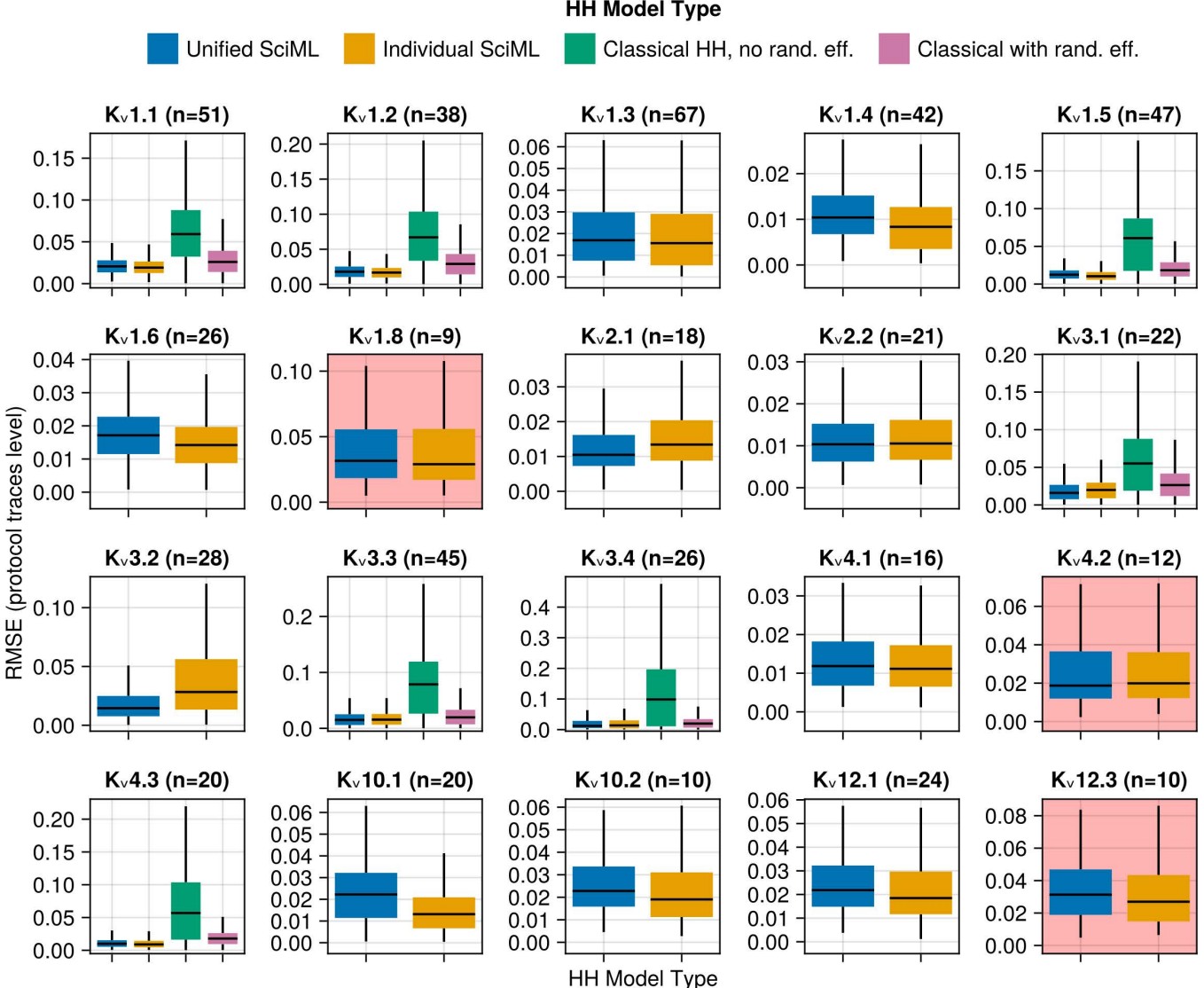

**Fig 4. RMSE box plots obtained by comparing the data against model predictions at the sweep level for all sweeps of all protocols, e.g., activation protocol at –10mV, deactivation at –40mV, inactivation at –30mV, etc. Different colours represent different HH model types: blue – unified SciML HH model, yellow – individual SciML HH models, green – classical HH models re-implemented from existing publications without random effects, pink – classical HH models re-implemented from existing publications, but with random effects on their parameters.** The number of different cells in the test data set is denoted by $n_{test}$ and shown above each subplot. The channel types ($K_v1.8$, $K_v4.2$ and $K_v12.3$) not used in the training of the unified SciML HH model have a light red background to distinguish them from the rest. Note that outliers are not shown and the $y$ axes are not identical between different subplots to enhance legibility.

above the diagonal show the RMSE values associated with the pairs of EBEs, and could highlight regions of EBEs which would systematically have higher errors, but no such patterns emerged. We conclude that there is no significant mismatch between the random effect prior and the posterior.

The second set of diagnostic plots (Fig 8) are classical NLME residuals diagnostics. Fig 8A shows the relationship between the model population level predictions (i.e., not for a particular cell, but for a channel type) and all the observations for that channel type. An ideal model would have a Gaussian error around the yellow $y = x$ line, which denotes a

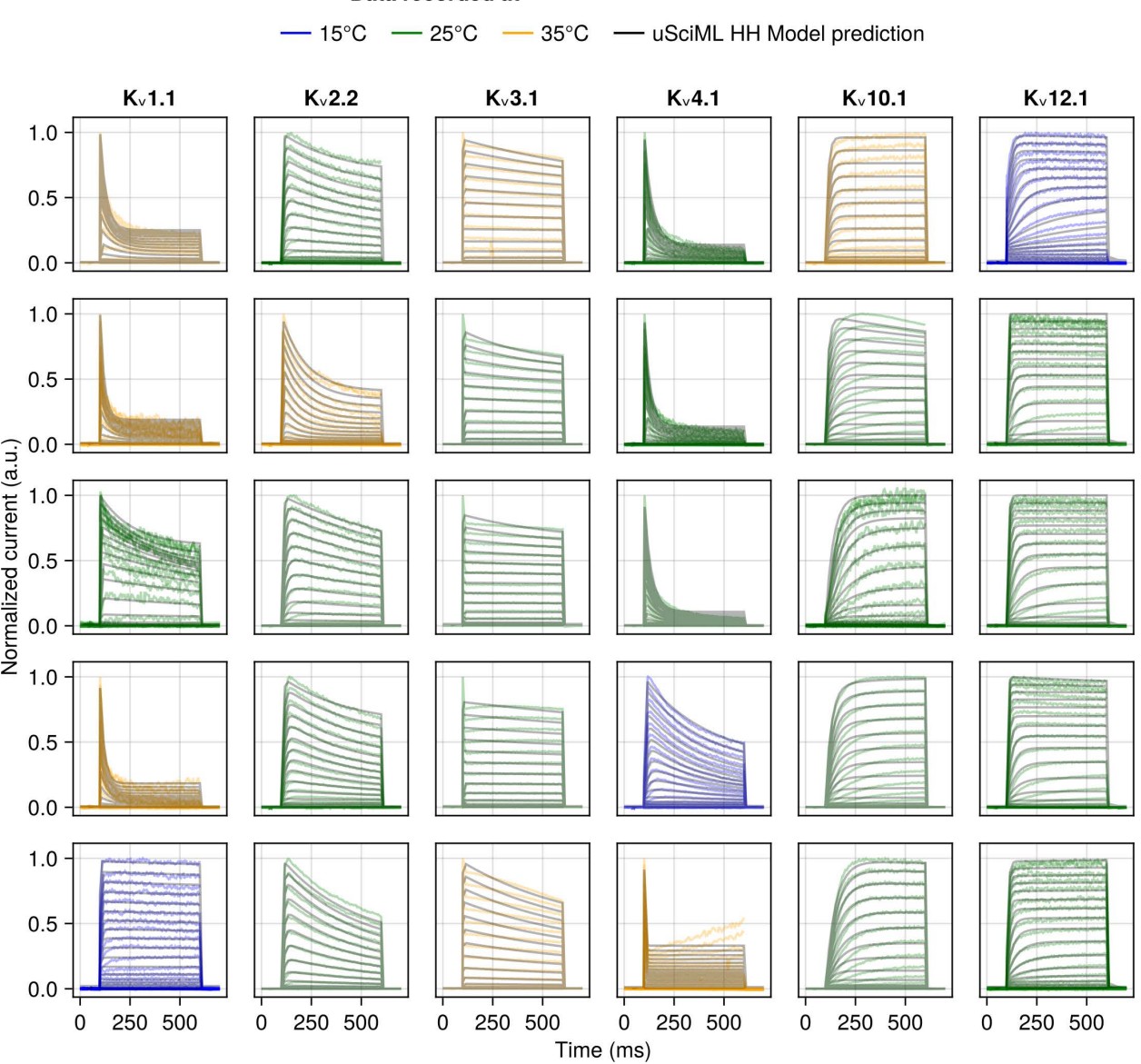

**Fig 5. Samples of the unified SciML HH model predictions for test data for the activation protocol for a subset of channel types used in training: $K_v1.1$, $K_v2.2$, $K_v3.1$, $K_v4.1$, $K_v10.1$ and $K_v12.1$.** The model predictions are plotted as black lines, whereas the empirical data is colour coded by the temperature at which it was recorded – blue for 15°C, green for 25°C and yellow for 35°C. Each subplot represents measurements from a different cell.

perfect relationship between the observed values and the values predicted by the model. The unified SciML HH model shows approximately this behaviour as both the local linear fits to the contour lines (LOESS, red line) and the global linear fit (OLS, green line) are very close to the ideal line of $y=x$ (yellow line). The error distribution exhibits some heteroscedasticity – the width of the error distribution decreases with increasing $x$ values. However, overall, given that the heteroscedasticity is mild, we conclude that the model fits the data reasonably well at the population level.

Visually, the individual predictions (Fig 8B) are significantly more accurate than the population level predictions (Fig 8A). For the individual predictions, the global and local linear fits nearly coincide with the $y=x$ line and there is a high

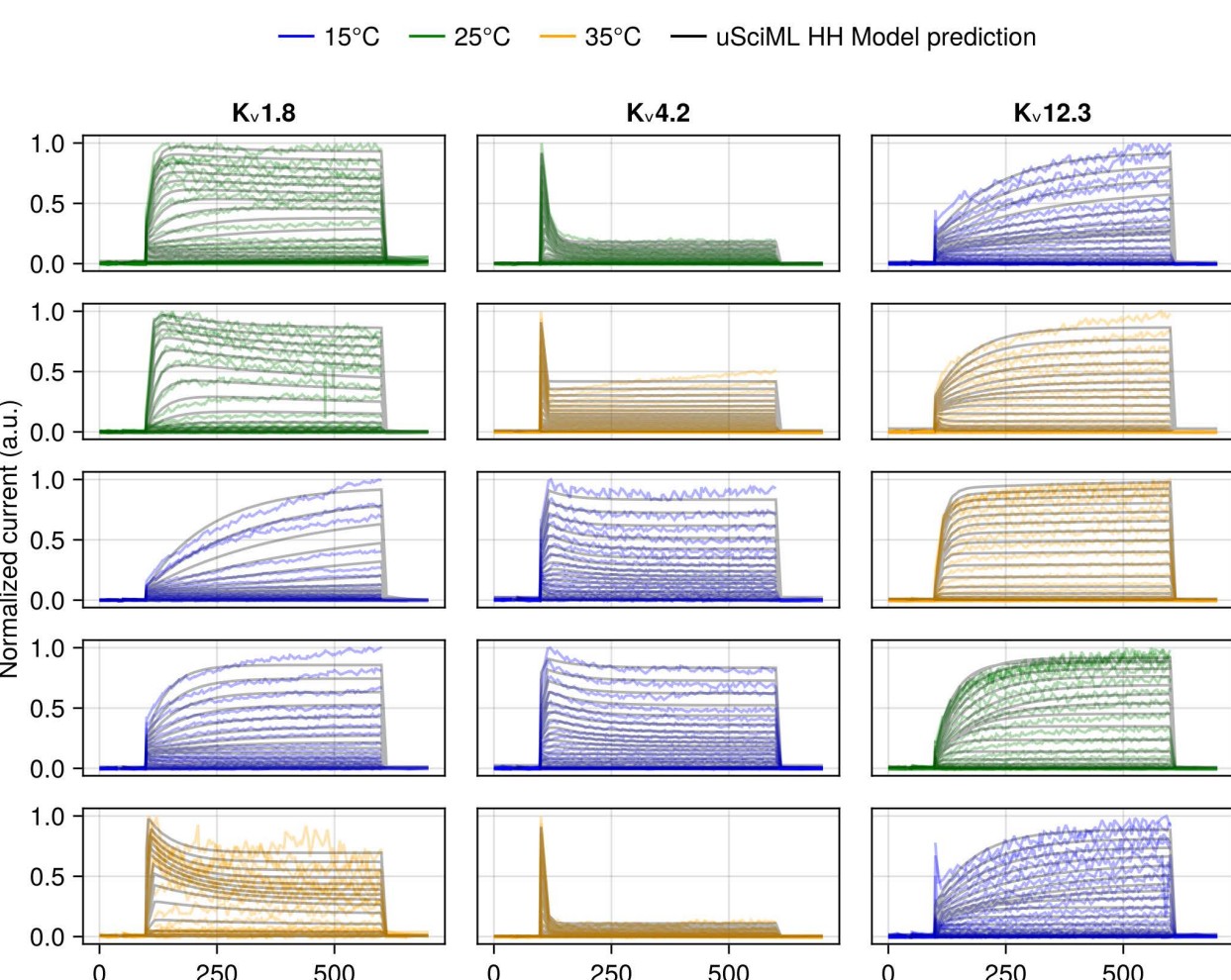

**Fig 6. Samples of the unified SciML HH model predictions for the activation protocol of the test data for a subset of channel types not used in training (K$_v$1.8, K$_v$4.2, K$_v$12.3) and therefore represent the model's ability to generalize.** The model predictions are plotted as black lines, whereas the empirical data is colour coded by the temperature at which it was recorded – blue for 15°C, green for 25°C and yellow for 35°C. Each subplot represents measurements from a different cell.

concentration of contour lines around the $y=x$ line with a relatively small standard deviation. Therefore, we conclude that the individual fits obtained via the unified SciML HH model are better than the population fits.

Fig 8C analyses the population level residuals weighted by the inverse standard deviation of the observation model and their dependence on time. Therefore, in this plot the residuals can be larger than 1 if a model has a small $\sigma$ value in the observational model. A well-performing model would show no dependence of the residuals on time, which is nearly true of the unified SciML HH model – the local and the global linear fits (red and green lines) are virtually overlapping and are on the $y=0$ line. The wider distribution of the contour plot of the residuals at earlier times might indicate that there is a higher spread of the residuals during those times. However, the length of the recording from a cell was proportional to the number of stimulation protocols and sweeps which were of high enough quality. Therefore, since a significant amount of data was filtered out, there are more shorter recordings than longer ones, and the more numerous shorter recordings will tend

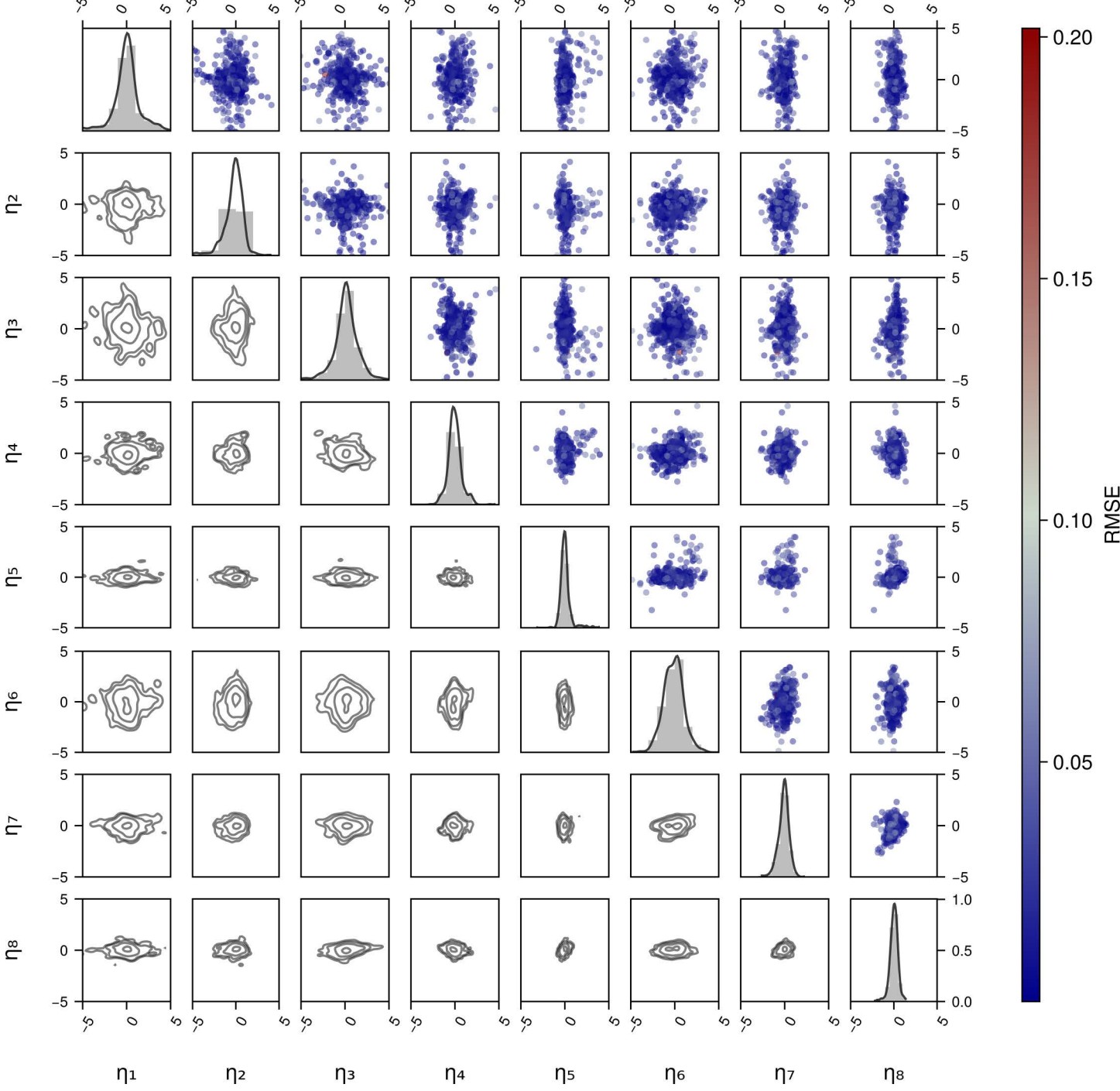

**Fig 7. Pair plots of the Empirical Bayes estimates (EBEs) of random effects obtained on the cells in the test data set.** The diagonal shows the histogram and a kernel density fit to the EBE distributions. Sub-diagonal subplots show the contour plots of the distribution of pairs of EBEs. Super-diagonal subplots show the scatter plot of the pairs of EBEs with points coloured by the RMSE at the sweep level to check if certain regions of the distributions contain high RMSE points.

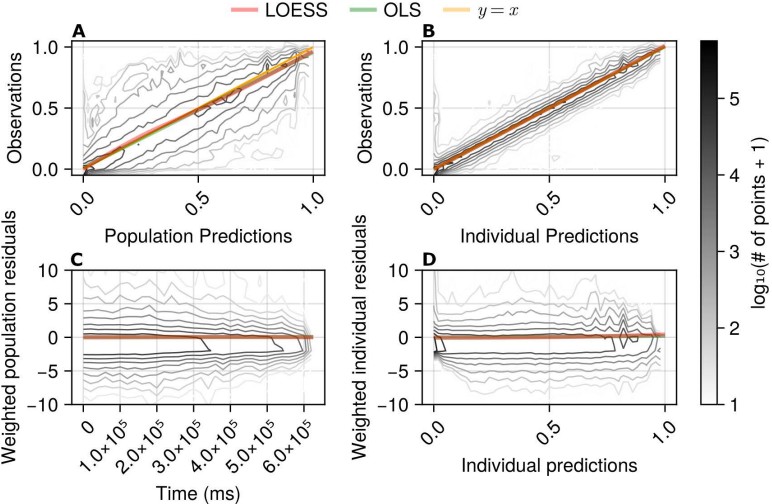

**Fig 8. Goodness of fit visualizations for the unified SciML HH model on the test data set, showing: (A) observations plotted against population predictions; (B) observations plotted against individual predictions; (C) weighted population residuals plotted against time; and (D) weighted individual residuals plotted against individual predictions.** Each subplot contains either two or three lines: the yellow $y = x$ line (where $x$ is the prediction and $y$ are the observed values), in the top two plots shows an ideal scenario, the green line represents a global ordinary least squares (OLS) fit and the red line represents the locally estimated scatter plot smoothing (LOESS) fit to the data shown in each subplot. The contour line levels represent the $\log_{10}(\text{\# of points} + 1)$ (differentiated by the intensity of the contour and shown in the colour bar on the right) and are based on approximately 3 million observed data points present in the test data set.

display larger variability than the fewer long ones. Therefore, the higher spread of the contour lines at earlier times is most likely a feature of the data used, rather than biased model performance.

Finally, Fig 8D plots individual predictions against weighted residuals of individual predictions. Under ideal circumstances the plot would show a $\mathcal{N}(0, 1)$ distribution around the $y = 0$ line. The model results show the correct mean behaviour both locally and globally (red and green lines basically overlapping), but not ideal spread that is skewed towards the negative values with a standard deviation larger than one (similar skewness can also be noted in the bottom left plot). This skewness is most likely due to the model structure and normalization of the data. The model cannot produce negative normalized currents, whereas, due to the noise inherent in the data, current values can be negative and they get up-weighted by a small $\sigma$ in the observational model. Therefore, these diagnostics support the conclusion that the unified SciML model predictions are generally unbiased, both at the population and the individual cell levels.

The final set of diagnostics are the visual predictive checks (VPCs) shown in Fig 9. The VPCs are given for each sweep of each protocol to provide a more granular diagnostic of the model's performance. For a given model VPCs a produced by sampling $k$ random effects which produces $n$ populations. For each population of $k$ samples we calculate the 95% confidence intervals for the 10%, 50% and 90% quantiles of the data (shaded areas) and then plot the actual observed quantiles. If the observed quantiles fall into the simulated 95% CI of the quantiles, the model is representative of the data to which it was fit.

Overall, the VPCs support the conclusion that the unified SciML HH model fits the population level data reasonably well. However, most likely due to the differing amounts of data available for each protocol, there are some differences between the protocols. More specifically, the activation and deactivation protocols show mostly good VPCs, whereas the inactivation and the recovery protocols show still mostly good, but somewhat worse VPCs. Specifically, for the inactivation protocol the model consistently under-predicts the 10% quantile relative to the data. Similarly for the recovery protocol, the model systematically under-predicts the 10% quantile. Therefore, the VPCs indicate that some caution is warranted when

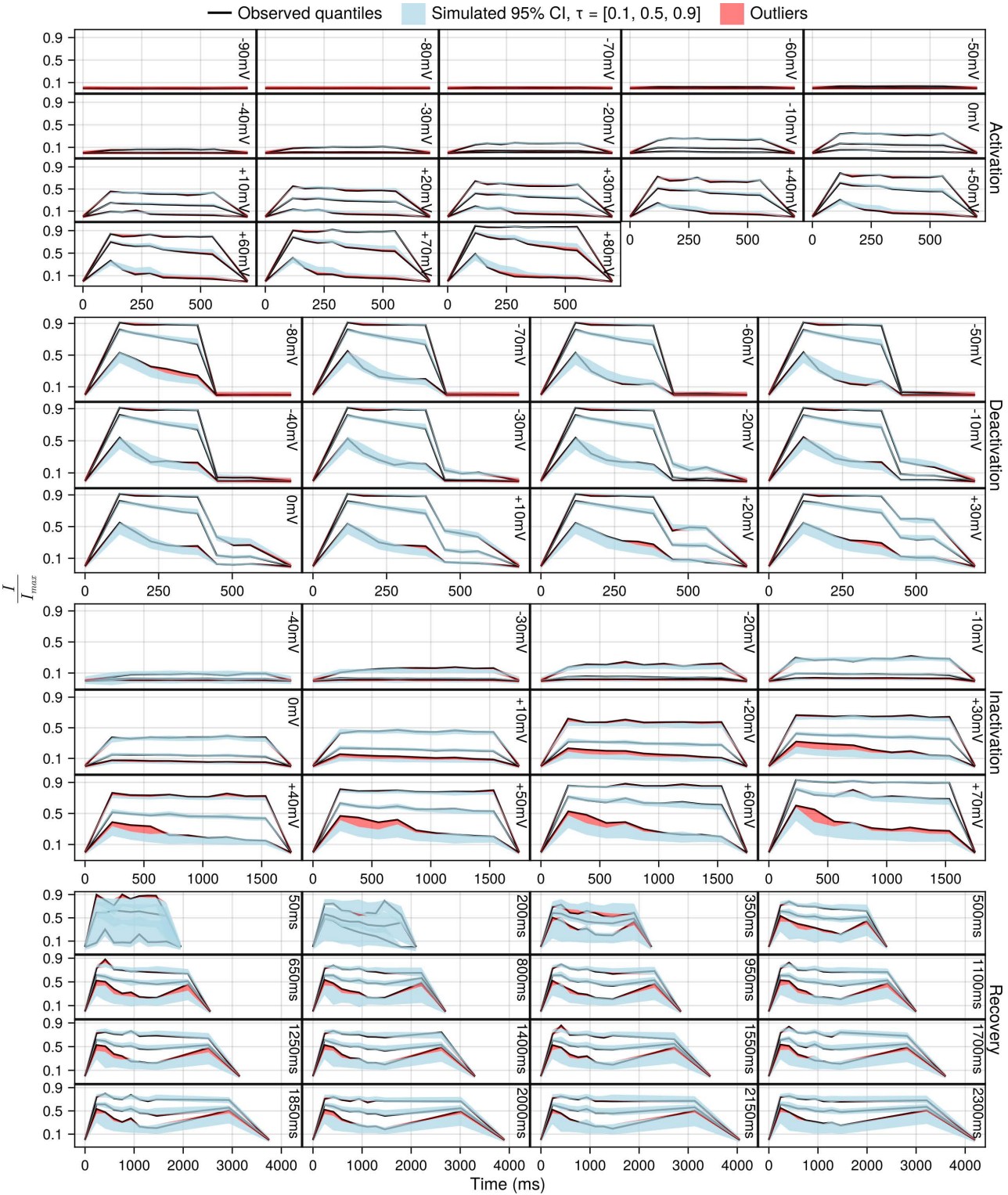

**Fig 9. Visual predictive checks (VPCs) for the unified SciML HH model with one subplot for each sweep of each protocol.** Each subplot contains three black lines representing the 10%, 50%, 90% quantiles of the data, as well as three light blue shaded areas for the simulated 95% confidence intervals for the 10%, 50%, 90% quantiles. The red shaded areas show where the observed quantile lines are outside the simulated quantiles.

the unified SciML HH model is used to simulate channel dynamics involving inactivation or recovery, but otherwise provides general support to the quality of the population predictions of the model.

We have provided three model diagnostic plots for the unified SciML HH model, investigating different, if overlapping, aspects of its performance: whether the EBEs are correlated, the residuals and the VPCs. The diagnostics lend significant support to the conclusion that the model predictions for individual cells are highly accurate and the population level predictions are reasonably accurate.

**Unified SciML Hodgkin-Huxley model gating function behaviour**

Having established the quality of the unified SciML HH model, we now investigate its properties, starting by showing the fitted $m_{i,\infty}(V, T)$ and $\tau_i(V, T)$ functions. We plot the fitted functions for the individual cells for the test data set in Fig 10 and the population predictions in Fig 11. We provide both plots to illustrate the variability of the gating functions in the test data, and to show the population level predictions made for temperatures not included in the training data.

The individual fitted functions (Fig 10) exhibit several patterns. Firstly, $m_{2,\infty}(V, T)$ approximately follows the classical sigmoidal behaviour of an activation gate, increasing in value as the voltage increases, reaching the maximal value of 1 at different voltages for different $K_v$ types. It is notable that $m_{2,\infty}(V, T)$ shows a small amount of temperature dependence.

The steady state behaviour of the first gating particle, $m_{1,\infty}(V, T)$, however, varies between cells, displaying three clusters of behaviours. Firstly, for some channel types ($K_v10.1$, $K_v10.2$, $K_v12.1$ and $K_v12.3$) $m_{1,\infty}(V, T)$ does not show significant dynamics, implying that for those channel types a single gating variable would have been sufficient. Next, a few channel types display a classical inactivating behaviour that may reach close to full ($K_v1.3$ and $K_v3.4$) or partial inactivation ($K_v1.5$, $K_v3.1$ and $K_v4.3$). For all remaining channels, $m_{1,\infty}(V, T)$ shows a mix of inactivating behaviour at low voltages and activating behaviour at high voltages (e.g., see $K_v1.8$ or $K_v4.2$, especially at 15°C). This behaviour illustrates the flexibility of the SciML approach. This behaviour may also indicate that adding a third gating variable may provide small improvements for the channel types showing this behaviour in $m_{1,\infty}(V, T)$.

Contrasting $\tau_1(V, T)$, the time constant associated with $m_{1,\infty}(V, T)$, and $\tau_2(V, T)$, the time constant associated with $m_{2,\infty}(V, T)$, it is clear that $\tau_1(V, T)$ is generally either only as fast or significantly slower than $\tau_2(V, T)$. Therefore, the main activation represented by $m_{2,\infty}(V, T)$ is fast, and the gating dynamics modelled via $m_{1,\infty}(V, T)$ are slow. Generally the $\tau_i(V, T)$ behave as expected when temperature is increased, in they get smaller (i.e., the equilibration is faster). Recall that $\tau_i(V, T) = \frac{1}{\alpha_i(V,T)+\beta_i(V,T)}$ – if the temperature is increased, reaction rates $\alpha_i(V, T)$ and $\beta_i(V, T)$ tend to increase, decreasing the time constant $\tau_i(V, T)$.

Contrary to the results shown in Fig 10 which are based on actual fits using $\eta$ obtained from individual cells in the test data set, the plots in Fig 11 are obtained by setting $\eta = 0$ and setting temperature $T \in [15, 20, 25, 30, 35]$. Therefore, Fig 11 provides predictions in cases where no data is available, specifically at temperatures other than 15°C, 25°C and 15°C. For both $m_{1,\infty}(V, T)$ and $m_{2,\infty}(V, T)$ their dynamics across temperatures are monotonic, suggesting that the model may generalize reasonably to other temperatures. It is notable that $\tau_1(V, T)$ can take various different functional forms (e.g., see $K_v1.8$ or $K_v3.4$) across temperatures. Finally, $\tau_2(V, T)$ shows a fairly consistent functional shape with the largest differences between channel types at higher temperatures.

Finally, Fig 12 shows the $Q_{10}(V)$ values for ratios of the gating time constant functions evaluated at 15°C and 25°C (blue lines), and the ratios of functions evaluated at 25°C and 35°C (orange lines). The most striking feature of these plots is the $Q_{10}(V)$ values taken for the activation time constant $\tau_2(V, T)$, e.g., 100 at +80mV for $K_v3.3$ and smaller (but still large) for other channel types. These $Q_{10}(V)$ values significantly exceed any previous $Q_{10}$ values (e.g., 2–4, see [1]) used in models of ion channels, as well as empirical estimates provided in [5]. The $Q_{10}(V)$ values for $\tau_1(V, T)$ are better aligned with the literature. Importantly, most channel types show significant voltage dependencies for $Q_{10}(V)$ that can be qualitatively different between the two calculated ratios. Data measured at a set of intermediate temperatures between 15°C and 35°C may be necessary to better constrain the $Q_{10}(V)$ values of the unified SciML HH model.

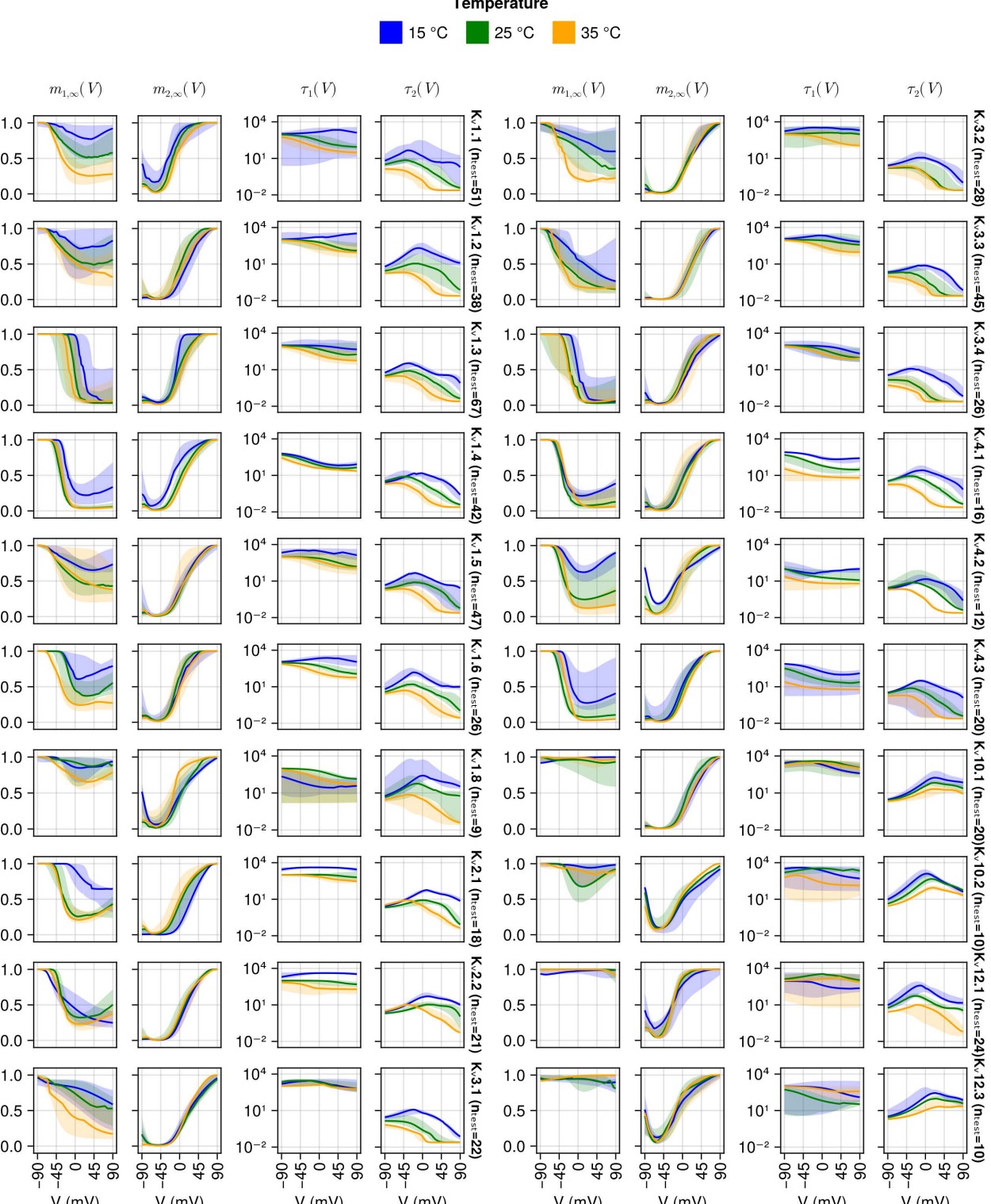

**Fig 10. Individual predictions of the $m_{i,\infty}(V, T)$ and $\tau_i(V, T)$ functions with the EBEs for the cells in the test data.** Each column represents one plots one function and its dependence on voltage and the plot is split into two groups, the left batch of $K_v$ channels and the right batch of $K_v$ channels, indicated by the names after the fourth and the eighth columns with the number of cells $n$ in the test data for that channel type. Most cells had data in all three temperatures – 15°C (blue), 25°C (green) and 35°C (yellow). The solid lines represent the median and the shaded area is the 95% confidence interval.

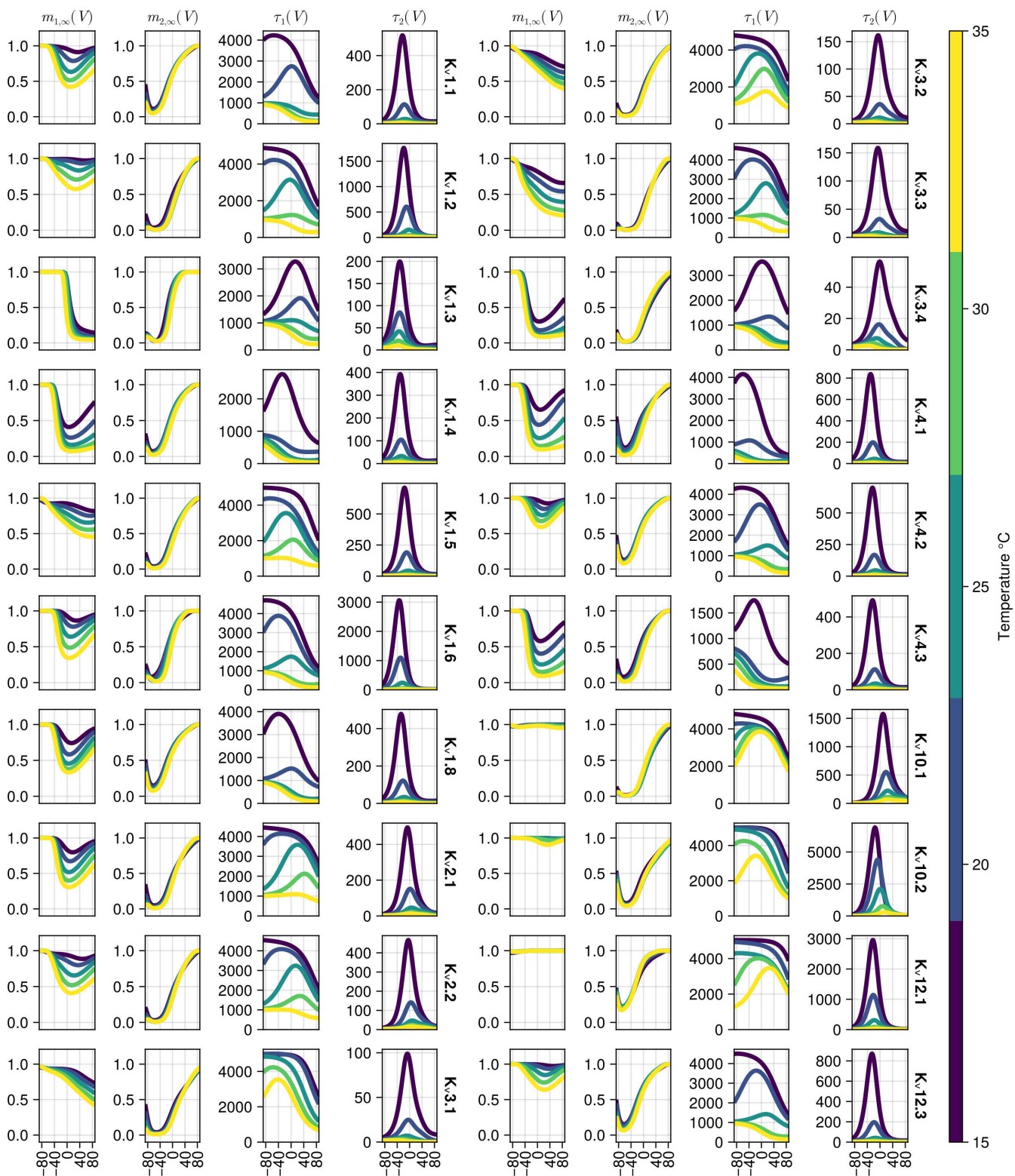

**Fig 11. Population predictions of the $m_{i,\infty}(V, T)$ and $\tau_i(V, T)$ functions with the $\eta = 0$.** Each column represents plots for one function and its dependence on voltage and the plot is split into two groups, the left batch of $K_v$ channels and the right batch of $K_v$ channels, indicated by the names after the fourth and the eighth columns. The colour bar on the right indicates the temperature at which the functions are evaluated.

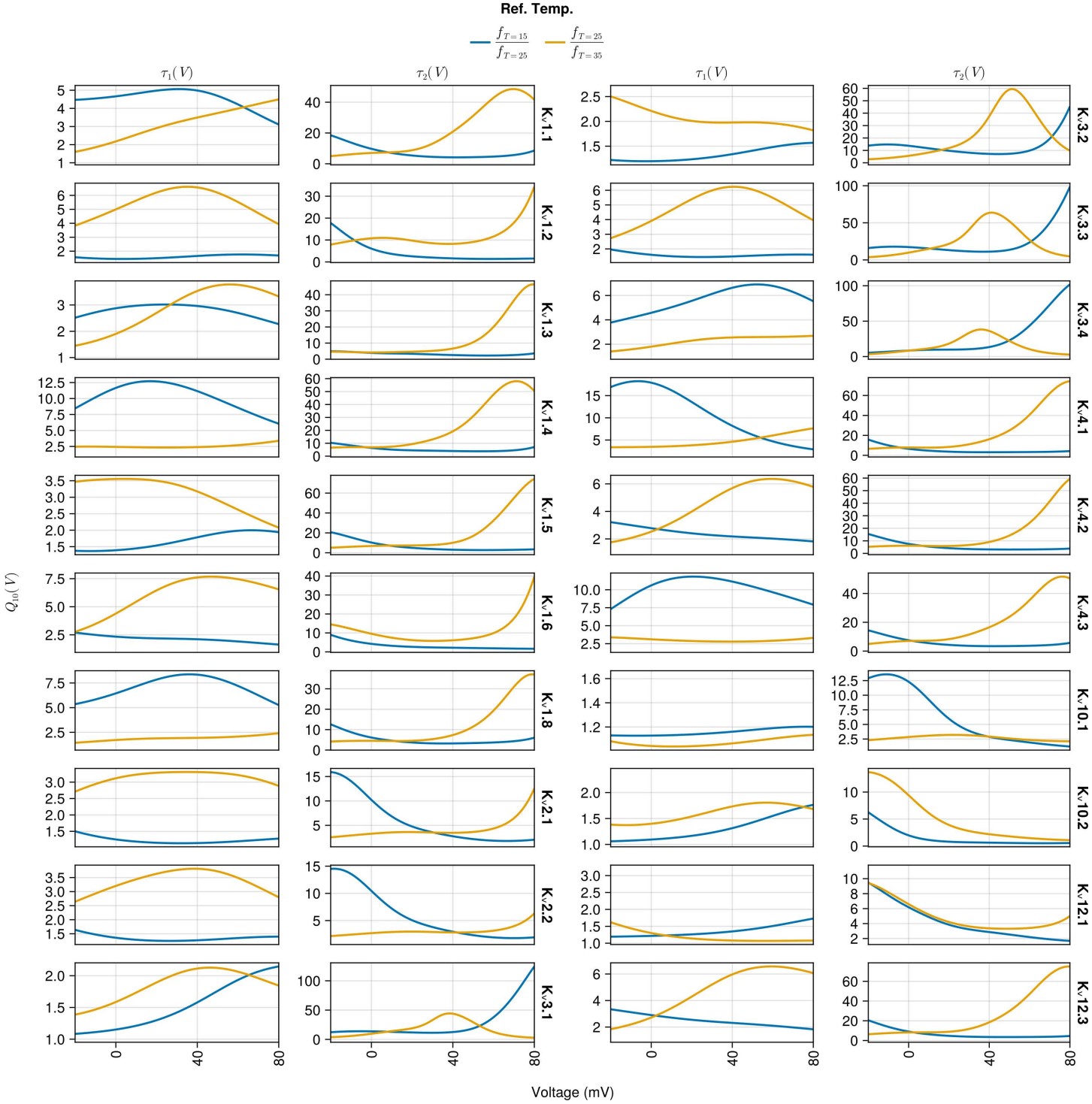

**Fig 12. $Q_{10}(V)$ values obtained by evaluating the unified SciML HH model with $\eta = 0$ and calculating the ratios of the gating time constant functions $\tau_1(V, T)$, $\tau_2(V, T)$ values evaluated at 15°C and 25°C (blue line) as well as 25°C and 35°C (orange lines).** The plot is split into two blocks (two columns each) and each row within a block is for an individual $K_v$ channel type.

## Discussion

In this study, we employed scientific machine learning (SciML) and non-linear mixed effects modelling (NLME) to address the challenge of modelling the gating kinetics of a diverse set of voltage-gated potassium ion ($K_v$) channels, using data from Ranjan et al. [5]. To our knowledge, no previous work has modelled this data set fully, in a way that would fit individual cell recordings for a large number of cells and the majority of the recording protocols. The features of the traditional Hodgkin-Huxley-like (HH) models, such as the appropriate number of gating particles, the functional forms and parameters describing the voltage dependencies of these particles, the exponents applied to the gating particles, are informed by a mix of assumptions, optimization techniques, as well as biophysical theory and observations including protein structure, where known. In contrast, we utilized the SciML framework, replacing the traditional functional forms of $m_{i,\infty}(V)$ and $\tau_i(V)$ with neural networks that were appropriately constrained, relying on the constraints of biophysical theory to a significantly lesser degree. This allowed us to preserve the general HH framework while only assuming the number of gating particles. All other parameters were then fitted using efficient, gradient-based optimization techniques [28]. The functional forms discovered via the data-driven approach, given their usefulness in fitting the data, may serve as an inspiration to develop new theoretical approaches in future studies.

The data we fit the models to showed significant heterogeneity [5]. In addition to the expected differences in gating kinetics between distinct $K_v$ channel types, variability was also observed within cells expressing the same $K_v$ type. This within-type variability is challenging to capture with conventional methods for modelling voltage-gated ion channels. However, the NLME approach is specifically designed to take into account both the variability shared between the cells and unique to them, by modelling the latter via random effects.

We explored two distinct SciML HH modelling approaches: (i) constructing individual SciML HH models for each $K_v$ type, and (ii) developing a unified SciML HH model that accommodates all $K_v$ types for which there was data, with channel type mapped onto the random effects through an additional neural network $NN_{aug}$. To assess the relative performance of the SciML HH $K_v$ models, we compared them to a set of seven previously published HH models. Since the existing HH models do not account for within-$K_v$-type variability, we incorporated random effects into the parameters of the seven published $K_v$ channels. This enabled us to focus more on the contributions of the SciML approach – i.e., the data-driven learning of voltage-dependent functions – and evaluate the impact of incorporating random effects in fitting the Ranjan et al. [5] dataset.

Our comparisons demonstrated that both the addition of random effects and the use of the SciML approach independently improved the fit of the $K_v$ models to the data (see Fig 4). The seven existing HH models fit significantly better when random effects were added. Furthermore, the HH models with random effects were significantly outperformed by the two SciML HH models, in which neural networks represented $m_{i,\infty}(V)$ and $\tau_i(V)$. The next question was whether the individual or unified SciML HH approach performed better. Comparing the two approaches revealed that the individual SciML HH models generally outperformed the unified model. This result is expected, as the neural network architectures used in both approaches were identical. The individual models could specialize more effectively to the specific gating dynamics of each $K_v$ type, whereas the unified model had to account for all types. However, while the differences between the individual and unified models were statistically significant, they were generally small. Notably, the unified SciML HH model was able to generalize to three additional $K_v$ types that it was not trained on. Given the small performance differences and the advantage of parsimony, we concluded that the unified SciML HH model is preferable. It is also highly extensible to many more voltage-gated channel types. Therefore, our primary contribution is the development, validation, and analysis of the unified SciML HH model, capable of modelling the gating dynamics of at least 20 different $K_v$ channel types.

The creation of a unified model for 20 different $K_v$ channels represents an important first step, but there are many potential directions for future development. The most immediate and promising future direction is extending the current unified model to include recently published data [5] on other major voltage-gated ion channels ($K_{2P}$, $K_{ir}$, $K_{Ca}$, $Na_v$, $Ca_v$, and HCN). While some families of voltage-gated channels, such as Kir and Nav, should integrate smoothly into the current

unified model – potentially even falling within its existing predictive range – other families with more complex gating mechanisms (e.g., $K_{Ca}$ and $Ca_v$) will require more sophisticated adaptations. For these, additional components, such as calcium dynamics, may need to be incorporated into the model. Moreover, there is no theoretical limit to the number of temporally varying signals (such as neuromodulators, drugs, etc.) that could be fed as inputs to the neural networks representing the gating functions, depending on the available data. However, practical challenges arise when adding more dimensions or inputs to the neural networks, and alternative approaches would need to be explored to assess their feasibility.

One of the potential challenges for the unified SciML HH model lies in its use of one-hot encoding to represent different channel types. If this encoding is naively extended by increasing its dimension, it may lead to an augmentation neural network $NN_{aug}$ with too many parameters, making the model prohibitively expensive to fit. A more practical long-term solution could involve a dimensionality reduction approach, perhaps based on factors like the 3D structure of the channel or its amino acid sequence. The latent representations derived from such an encoding could then be combined with the random effects or used to replace the current one-hot vector altogether. This approach could enable the model to predict the impact of point mutations or other modifications on channel kinetics. Furthermore, encoding the 3D structure or amino acid sequence would facilitate the creation of species-agnostic ion channel models, allowing for the integration of data from different species. A similar dimensionality reduction approach could be applied to external factors affecting channel kinetics, such as neuromodulators or drugs, before passing these latent representations into the neural networks that model the kinetics. In essence, the use of a foundation model – currently popular in deep learning – could be highly beneficial for modelling ion channel gating [43]. It remains to be seen what predictive and explanatory power such a model could offer.

The HH approach to modelling ion channels – where channels are represented by a number of independent gating particles – is just one of several modelling paradigms. In fact, it is a subclass of the broader Markov models, where the interaction of gating particles in determining the open channel fraction may be more complex than a simple product. For additional modelling approaches, see [44]. Recent studies have used single-channel patch-clamp recordings combined with deep learning to infer the most likely Markov schemes underlying the recorded data [12,45]. Although their approaches may initially appear similar to ours, there are several key differences. Firstly, the studies by [12,45] rely on single-channel patch-clamp recordings, which are more challenging to obtain compared to the whole-cell patch-clamp data used in Ranjan et al. [5]. Consequently, these approaches are suited to different types of questions. For instance, if the focus is on understanding the impact of phosphorylation on the gating of individual channels, the approach taken in [12,45] would likely be more appropriate. However, if the goal is to model the aggregate dynamics of many channels within a large patch of membrane, our approach is likely to provide more accurate predictions. Moreover, our approach is better equipped to handle variability from different sources, such as differences in gating kinetics between cells expressing the same $K_v$ type or across different $K_v$ types. Additionally, extending the gating mechanisms to include $Ca^{2+}$ or other ligands, in addition to voltage, may present more challenges for the Markov scheme approach due to the exponential growth in the number of states. Theoretically we could take different Markov schemes and represent the voltage-dependent gating functions as neural networks, but such an approach would require significant efforts to find an appropriate Markov scheme. Therefore, the approaches in [12,45] and ours may be complementary. If the HH formalism proves insufficiently expressive for modelling a broader range of ion channel types, the approach of [12,45] could provide a more universally appropriate Markov scheme. Our approach could then be applied to model the transition rates within it.

In addition to its explanatory and predictive benefits, a foundational approach to modelling ion channel gating could offer practical advantages for modellers. Current detailed compartmental models of neurons (written in, e.g., `NEURON`) typically require multiple files, each representing a single type of ion channel. There is an abundance of such files in various repositories and databases (notably https://modeldb.science/ [8]), but these files often differ in their underlying models, even for the same ion channel type. Determining which model is most appropriate and provides the best fit to the data can be difficult and time-consuming. However, if a well-validated foundational model were available, this problem would

 

be significantly alleviated. The burden of creating and maintaining a large number of files for each ion channel would be greatly reduced. Our unified SciML HH model could currently be used in a `NEURON.MOD` file by incorporating a pre-constructed lookup table for different $K_v$ types. However, a significant challenge arises from the inability to use multiple lookup tables within the same `.MOD` file. Other data formats, such as NeuroML [46], may present their own challenges when attempting to port our unified SciML HH model to them. If foundation channel gating models are developed in the future, it would be important to collaborate with software developers of compartmental neural modelling simulators.

There are multiple ways to fit HH models to current recordings and it is worth comparing the software used in this study (`Pumas.jl` and `DeepPumas.jl`) to other available software packages. For example, `Data2Dynamics` (https://github.com/Data2Dynamics/d2d) is a MATLAB-based software package capable of optimizing the reaction rates of chemical reaction networks in an efficient manner [47]. This software suite was used in an attempt (https://github.com/njohner/Kv-kinetic-models) to model the $K_v$ data from Ranjan et al [28]. Even though `Data2Dynamics` offers powerful capabilities in optimizing the reaction rates of chemical reaction networks, it does not include algorithms for the optimization of the structure of the networks. Therefore, while powerful in certain use cases, the problem tackled in this study requires a richer set of functionalities, notably usage of neural networks. Another option that is more specialized to computational neuroscience is the Python-based `BluePyOpt` package [48]. It has been used in studies to optimize the conductances of ion channels along the dendritic tree of a compartmental neuron model [49,50]. It uses a set of evolutionary algorithms to fit various parameters, such as the maximal channel conductance, distribution of conductances along a dendrite, but it can also be used to optimize the parameters that determine the kinetic properties of individual channel types. However, `BluePyOpt` does not directly optimize the structure of the functions used for channel gating and therefore does not offer all the functionality necessary to tackle the current challenge. Without going into an extended review of the available tools for fitting chemical reaction networks to data, based on the two aforementioned examples, we conclude that `Pumas.jl` and `DeepPumas.jl` offer some of the currently most powerful and feature complete tools and were essential in modelling the different types of $K_v$ channels. Therefore, our second contribution is the application of a novel set of approaches, SciML and NLME, to a challenge in computational neuroscience. These tools can be applied to other challenges where significant variance between observed entities is present, for example, modelling of synaptic signalling stemming from highly variable proteomic structures [51].

In conclusion, we successfully addressed a major challenge in neuroscience: the efficient and accurate modelling of voltage-gated potassium channel gating. By combining two novel approaches in computational neuroscience – scientific machine learning and non-linear mixed effects modelling – we developed a unified model capable of fitting the gating dynamics of 20 different $K_v$ types. Our work demonstrates how these approaches can be seamlessly integrated into existing computational neuroscience approaches. Moreover, the produced unified $K_v$ channel gating model may serve as the first step in creating a next generation foundation model that would be able to model all known voltage-gated channel families. The resulting gains in predictive and explanatory power, as well as computational efficiency could help to tackle complex challenges, such as investigating diseases related to ion channels [52].

## Supporting information

**S1 Text. Hyper-parameter optimization.**
(PDF)

**S2 Text. Additional information about the baseline $K_v$ models used.**
(PDF)

**S1 Table. Channelpedia cell IDs used.**
(CSV)

## Author contributions

**Conceptualization:** Domas Linkevicius, Angus Chadwick, Melanie I. Stefan, David C. Sterratt.

**Data curation:** Domas Linkevicius.

**Investigation:** Domas Linkevicius.

**Methodology:** Domas Linkevicius.

**Software:** Domas Linkevicius.

**Supervision:** Angus Chadwick, Melanie I. Stefan, David C. Sterratt.

**Validation:** Domas Linkevicius.

**Visualization:** Domas Linkevicius, Melanie I. Stefan, David C. Sterratt.

**Writing – original draft:** Domas Linkevicius.

**Writing – review & editing:** Domas Linkevicius, Angus Chadwick, Melanie I. Stefan, David C. Sterratt.

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
