## [Decision Letter · Decision Letter 0]

7 Oct 2025

PCOMPBIOL-D-25-00808

One model to rule them all: unification of voltage-gated potassium channel models via deep non-linear mixed effects modelling

PLOS Computational Biology

Dear Dr. Linkevicius

Thank you for submitting your manuscript to PLOS Computational Biology. After careful consideration, we feel that it has merit but does not fully meet PLOS Computational Biology's publication criteria as it currently stands. Therefore, we invite you to submit a revised version of the manuscript that addresses the points raised during the review process.

Please submit your revised manuscript within 60 days (6th December 2025) If you will need more time than this to complete your revisions, please reply to this message or contact the journal office at ploscompbiol@plos.org. Please include the following items when submitting your revised manuscript:

We look forward to receiving your revised manuscript.

Kind regards,

Quentin Clairon, PhD

Guest Editor

PLOS Computational Biology

Hugues Berry

Section Editor

PLOS Computational Biology

**Journal Requirements:**

3) Please amend your detailed Financial Disclosure statement. This is published with the article. It must therefore be completed in full sentences and contain the exact wording you wish to be published.

2) If any authors received a salary from any of your funders, please state which authors and which funders..

**Reviewers' comments:**

Reviewer's Responses to Questions

**Comments to the Authors:**

Reviewer #1: 0. After the Introduction and up to the Methods section, the authors only mention what they do,

but they don't explain why or from what references they draw their writing.

This is particularly true of the parameters chosen, the methods used, and the general sequence

of the process and the document. Some explanations are vague.

It would be ok if some additional figures could be included to be clearer to the readers.

1. Please indicate the main characteristics of the data, to prevent the reader from obtaining the reference.

2. Minimum, maximum, average of cell recordings?

3. WHAT is the "sufficient" number of recordings ("sufficient" with respect to what and why?)

4. From (3): Why at those specific temperatures?

5. Authors provide and describe parameters with no rationale nor references. WHY do

you work exactly with those values?

6. Part: Data Processing. Authors claim that: "In order to make the data in Ranjan et al. [4]

suitable for model training we undertook 166

a series of data processing steps (Figure 1). First, we filtered out inconsistent data. We 167

then applied a time series smoothing algorithm to reduce the noise levels, followed by 168

setting the current baseline to 0, normalization of the current, rescaling, exclusion of 169

systematic data artifacts and downsampling".

Again, why those numbers and WHAT IS THE SMOOTHING algorithm? Why those parameters?

7. WHY this...?- All data processing, modelling and 170

analyses in this paper were performed using the Julia v1.10.4 programming 171

language.

8. Why to use MAPE and not other measure or more than 1 measure? Why 1 measure is ok and why only

one parameter is ok? How can you justify this?

9. Authors say that: "The dynamical parameters pn are then fed into the structural model (e.g. an ordinary 301

differential equation (ODE) system)". But where is the ODE system?

10. From (9), why not to use PDEs instead of ODEs? A good discretization could yield good results.

11. From (9,10), how do you know that the set of ODE actually have a solution?

12. Depending of the parameters of the ODE, the ODE may reflect some ill-posed issue, particularly with

small values of the coeffcients.

13. Line 308: What do you mean with : "the appropriate variables " ?

14. Eq. 4 implies a lot of things but you do not explain anything else.

15. How do ensure that your data has a joint Gaussian function?

16. About eq. (4), it has been proven that the uncertainty directly from Gaussian

Processes Regression is irrelevant to the cohort heterogeneity in general. What

happens in your case about this?

17. In real life, returns are not symmetrically distributed and exhibit extremes

not captured well by the bell-shaped curve of the Gaussian model.

18. Line 314: Maximum likelihood estimates can be heavily biased for small samples,

thus the optimality properties may not apply for small samples.

Maximum likelihood can be sensitive to the choice of starting value. What happens with yours?

19. The optimization problem should formally be posed.

20. How do you guarantee, in advance, that such optimum value will be found?

(global, local...).

21. How to be sure that the NN algorithm actually converges?

22. What is the computational and algorithmic complexity of the whole algorithm?

23. Why not to conduct a statistical test to test the classification results?

24. From (16-18): Unlike ML estimators, the MAP estimate is not invariant under

reparameterization. Switching from one parameterization to another involves

introducing a Jacobian that impacts on the location of the maximum.

In contrast, Bayesian posterior expectations are invariant under reparameterization.

Please comment abt this.

Reviewer #2: # One Model To Rule Them All

The authors present a set of hybrid models, derived from the individual and combined use of **Nonlinear Mixed-Effects (NLME)** and **Scientific Machine Learning**, to describe the **Hodgkin-Huxley (HH)** mechanism. These models are trained on real data from a dataset by Ranjan et al.

They provide a rigorous analysis of the results and classify their model among the existing models for gating mechanism modeling.

In my opinion, both the contribution and its representation are a well-founded innovation in the field of modeling these mechanisms. However, I believe there is a lack of an adequate introduction to the field of modeling in combination with machine learning models.

# Introduction

The authors provide a good introduction into the useage of HH models the challenges and the state of the art. However, one of the key contributions, namely the modelling using neural networks and its application in combination with first prior models is not metioned at all. Even though not many papers investigate these model types in combination with NLME models, its application in general scientific machine learning has been treated in various field. I advise that the authors give a brief overview on these developments to allow the reader to gain an overview on this topic.

# Methods

- p. 4, 138 is AP properly introduced somewhere ? If not, even if not used, I think it would be useful to get a further explanation on why this protocoll can be dropped. Is this due to the hope the neural network is able to account for this? Is this common practice ?

- p.5, 165 If the code for the data preprocessing available in the linked repository, consider mention the repo here.

- p. 11, 368 Why did the authors decide to use the same architecture for the individual and unified SciML model?

- p. 11, 352 In general the authors go through great length of introducing methodoly used. However, I find that the information on inputs and outputs of the neural network(s) is a little hard to find for such a prominent part of the paper. Consider to use proper equations instead of inlining here.

# Results

- p. 14, Fig. 3 I am honestly surprised to see the performance of the classical model structure over the data. Even without a direct comparision I would not use this model. Are there more elaborate, first principle model better explaining the data ?

- p. 19, 501 ff. The conclusion drawn for the individual and unified model are correct. However, due to the training the expected benefit might vanish, given that the individual models are not able to specialize.

- p. 22, Fig. 8 I really like the style of these goodness of fit visualizations, which where new to me with the contour plots. However, I find it challenging to visualize the implication quickly due to the choice of a the colorscale. The authors might choose to switch to a colorized colorscale for the derivation and indicate the LOESS, OLS and identity relation (which overlay anyway ) using black, red, or white ?

# Discussion

- p. 27 (There are multiple ways...) While the additional comparision between the different software packages is certainly informative, I propose that the authors move this comparision a) into section METHODS and maybe add a tabular summary or b) they move it to the back of the discussion given that it seems to divide two parts concerning the investigated models.

Reviewer #3: The authors present a new computational pipeline for modeling and fitting voltage gated potassium channels which they juxtapose against Hodgkin Huxley style models. Here the authors combine machine learning optimization techniques with non-linear mixed effects models to capture and fit the kinetic properties of these channels. The authors claim (and successfully show!) that this modeling paradigm is perfect for capturing the breadth of heterogeneity observed within a single Kv channel type. Their model does an excellent job at replicating/fitting the data taken from an online data repository. Specifically this model is able to easily capture the between cell heterogeneity that is observed in the data in a better manner than some previously published HH models. The biggest novel takeaway from the study is the exciting and novel application of machine learning and non-linear mixed effects modeling to study gating dynamics. The final conclusion is that with the power of this modeling technique, all Kv channel types can be reasonably modeled/fit with an HH style model that has 2 (complex) gating equations. Overall this is a nice presentation of a new (to this application) modeling framework being applied to the complex and noisy data produced by ion channels, and presents a useful and usable tool for future modelers.

One component of the paper that should be expanded and clarified/discussed further is their characterization (or at least the presentation) of the perceived issues and applicability of HH and Markov modeling. Namely, when the authors discuss the weaknesses of HH/Markov models they make claims about these models exhibiting numerical instability, exhibiting extreme behaviors, and requiring expensive optimization routines or manual hand-tuning . These claims appear to be lacking citations and a clear justification. For HH numerical/instability issues: which models (and functional forms) explicitly are being discussed here (or are they claiming that all HH models suffer from these issues which would be incorrect)? For optimization: here the authors are using a complex fitting algorithm/tool (ML) to fit their models while employing multiple CPUs and taking multiple hours, and it is unclear from these statements if current HH optimization/fitting is worse (more expensive) than this using tools like BluePyOpt (mentioned by authors). Then, regarding the comparisons between the new SciML model and HH models, the comparisons between performances on the data appear incomplete (see comments under Major revisions). Finally, their discussion of traditional HH/markov models is somewhat misleading to readers in talking about assumptions about the number of gating particles and functional forms for these models. In the case of (well executed) HH/markov modeling works, these assumptions are derived from the biophysics and statistical mechanics that govern the activating and inactivating variables and their functional forms. These “assumptions” (at least about the number of gates, or the Markov structure as a whole) enable the modeler to generate predictions (or are based on knowledge) about how the ion channel is functioning at the protein level. Instead of framing this as a weakness of HH models, the authors should consider how to compare their SCIML models with the biophysical properties that are encoded by HH/Markov models. Specifically, If having a model with 2 gates can do as well as a model with 4 gates (which matches the known number of protein subunits and the conformational steps being taken specifically by Kv channels), what predictions does this make about how the ion channel works at the protein level? Is this suggesting dependence between the conformational changes indicating that Markov models don’t work here? Do 4 state HH models do well at some temps but poorly at others suggesting that the channels structure and function is being altered by temperature? In summary, the authors treatment of HH models seems not complete to justify some of the claims and analyses, at least for the compare and contrasting analyses being done.

Larger Revisions/Questions

Line 335-343 Are these preliminary tests shown somewhere? How is this initial quality of fitting to the data measured? If the models are described to be a reasonably good baseline, it should be justified how. It would also be nice/useful if the authors could include the HH models (at least in the supplement) for readers to compare them with the provided NLME models generated by the authors.

(Figure 3 and Figure 4). In tandem with the previous comment, why is this particular classical HH model chosen/utilized here? Can these HH model parameters be refitted to the data provided here (keeping the identical structure but refitting the rate constants)? Or at least refit to an average of the data? The HH models seem to perform so poorly that their initial structure itself (i.e. the number of gates or types of gates) is incorrect based on the data provided. That is to say, if a model without inactivation or deactivation kinetics built into the HH/Markov scheme was used to fit data that clearly exhibits this property, then the original model should not be considered an accurate/useful/good model. This indicates that comparing the SciML model to this particular HH model yields an unfair comparison of model types as clearly the SCiML model with an inactivation component will do better on data that inactivates than an HH model that is incapable of inactivating. What data were the HH models initially tested on such that they were deemed to be reasonable models (presumably at some point they fit some data well)? What temperature is this data at and at what temperature is the original data the model was fitted to? It seems that Kv1.1 channels exhibit inactivation at higher temperatures but not for lower temperatures. Is it possible that the original HH model is meant to be used for a lower temperature range (where there isn’t inactivation) but not at higher temperatures? In the very least, these comparisons should be done at other temperatures (or if there are multiple temperatures within this plot that should be made clear). Also more than just the response to a +80 voltage step should be shown. +80 reaches a voltage value that is non-physiologically relevant and thus might further exacerbate the poor performance of the unfit HH model. The summary plot in Figure 4 uses -10, and it is unclear why this wasn’t the value chosen for figure 3. Were comparisons between models conducted across all Voltage ranges?

Minor Revisions

Lines 35-36 Citations or more explanation would be helpful for the statements “exhibit extreme behaviors” and “numerically unstable”. Which functional forms are being discussed here from which papers? Is this in reference to the HH equations themselves, or just other models which have been fit using an m^xh^y paradigm. Are the equations utilized and referenced here as “unstable” properly derived from statistical physics as boltzmann equations consisting of standard exponential rate equations, or are they non-standard examples?

Lines 37-38 coupled with 106-109. Citations should be utilized here. What does expensive mean? The authors claim that this new model formulation is simpler, more efficient, fast and more powerful. It is unclear what metrics are used for defining and indicating that the techniques used to generate HH equations are slow. It is also unclear if this is discussing the formulation of the model itself, or the fitting process or both.

Line 123 “These channel types were chosen based on a preliminary inspection of the number of cell recordings that showed a good signal to noise ratio in a sufficient number of recordings over different temperatures ” What does a good signal to noise ratio in a sufficient number of recordings mean? This should be made precise.

Line 360 Authors should justify why i=2 gates was used. In the standard HH equations 4 gates are used due to the relationship with the protein composition and number of alpha protein subunits which undergo conformational changes to permit the passage of ions. N=4 was found to be used based on biophysical principles/mechanisms not arbitrarily. Here i=2 likely gives 1 activation and 1 inactivation gate type kinetics (which in fact is what the results end up showing), but it is unclear why 4 (to match Hodgkin Huxley and the protein subunit composition) or 1 (for simplicities sake) wasn’t chosen instead. An interesting speculation would be how adding additional gates would impact the instances where m1 has a mix of inactivating and activating behavior. Would this be broken up into two distinct gates?

Line 491 What do authors mean by “appropriate”. Does this simply mean “result in the best fit?”. What does constructed by hand mean? A citation would be helpful here to point out which types of models have biophysical support and which don’t. Which types of models are constructed by hand and which are derived from first principles? Are these model types being grouped together or considered separately?

Lines 660-664, Again the authors should emphasize or recognize that good HH models (albeit not all modeling which is claimed to be in the style of HH are good) are based on biophysical properties where the exponents and number of gating particles is determined from experimental data and the protein level structure. This is why the more general Markov structure (which reduces to the HH formalism) is useful in helping readers see the channel structure. Voltage clamp data enables experimentalists to identify what types of gates should exist (activation, inactivation), and then analyzing protein structure with imaging techniques helps determine the number of particles. This is no more of an assumption than the authors choosing to use 2 gating particles.

Line 632. A functional form of Ti is given here in terms of reaction rates, but these rates are not described earlier in the text. While these rate functions alpha and beta are consistent with HH modeling, it isn’t clear that the SciML framework is producing an alpha and beta term.

Figure 10, the scale on the Time constant plots makes it difficult for the readers to parse out how big some of these time constants are and thus how important this activation variable actually is.

Comments (these are largely just comments for speculation by the authors or things to consider including or pursuing in next steps)

Line 7 Non-specific or non-selective voltage gated channels (like HCN channels) are permeable to multiple ion types

Line 10 Take a look into heteromeric voltage gated potassium channels and the increased variability included here. Additionally there are beta protein subunits for KV channels that provide additional diversity in functional properties. These are simply both interesting things to think about moving forward for the Authors!

Discussion Comment: Which of the HH models used here have an n^4 structure? If so, how does the model with only two gates compare against this specifically? HH models using 4 gating particles match with the biophysics (4 protein subunits). But here the model uses only 2 gating particles no longer matching with the biophysical assumption of 4 subunits transitioning to the open conformational state. What does using two versus 4 gates say about the underlying biophysics? I’m merely interested in the authors’ ideas here and how it fits in with the known literature on gating particles.

Discussion comment: What is the author’s recommendation for what future users should do when attempting to include one of these channels in a cellular model? Should future users just use the population level models if they are unable to generate a fit for the individual cell type?

Results Comment: It is interesting that in some cases the inactivation time constant for M1 appears to be very large. This indicates that on the timescale that action potentials and neuron processing functions, M1 may be irrelevant. It’d be interesting to see how this aligns with the literature's descriptions of these channels’ kinetics and whether they are described as inactivating or not. Also is their a relationship between the scale of T1 and the shape of M1?

**Have the authors made all data and (if applicable) computational code underlying the findings in their manuscript fully available?**

The PLOS Data policy requires authors to make all data and code underlying the findings described in their manuscript fully available without restriction, with rare exception (please refer to the Data Availability Statement in the manuscript PDF file). The data and code should be provided as part of the manuscript or its supporting information, or deposited to a public repository. For example, in addition to summary statistics, the data points behind means, medians and variance measures should be available. If there are restrictions on publicly sharing data or code —e.g. participant privacy or use of data from a third party—those must be specified.requires authors to make all data and code underlying the findings described in their manuscript fully available without restriction, with rare exception (please refer to the Data Availability Statement in the manuscript PDF file). The data and code should be provided as part of the manuscript or its supporting information, or deposited to a public repository. For example, in addition to summary statistics, the data points behind means, medians and variance measures should be available. If there are restrictions on publicly sharing data or code —e.g. participant privacy or use of data from a third party—those must be specified.

Reviewer #1: **No:** Authors mention that they use specific software but without further clarification. Besides, explanations about how they created neural networks are missing. But what is quite remarkable is that the authors simply use parameters and equations in a narrative manner, without justification or bibliography. Although they use figures to show their results, it would be benefic to include diagrams, block diagrams to illustrate their processes.Authors mention that they use specific software but without further clarification. Besides, explanations about how they created neural networks are missing. But what is quite remarkable is that the authors simply use parameters and equations in a narrative manner, without justification or bibliography. Although they use figures to show their results, it would be benefic to include diagrams, block diagrams to illustrate their processes.

Reviewer #2: Yes

Reviewer #3: Yes

PLOS authors have the option to publish the peer review history of their article (what does this mean?). If published, this will include your full peer review and any attached files.). If published, this will include your full peer review and any attached files.

.

Reviewer #1: No

Reviewer #2: No

Reviewer #3: **Yes:** Kees McGahanKees McGahan

**Figure resubmission:**
---

## [Decision Letter · Decision Letter 1]

16 Jan 2026

PCOMPBIOL-D-25-00808R1

One model to rule them all: unification of voltage-gated potassium channel models via deep non-linear mixed effects modelling

PLOS Computational Biology

Dear Dr. Linkevicius,

Thank you for submitting your manuscript to PLOS Computational Biology. After careful consideration, we feel that it has merit but does not fully meet PLOS Computational Biology's publication criteria as it currently stands. Therefore, we invite you to submit a revised version of the manuscript that addresses the points raised during the review process.

We look forward to receiving your revised manuscript.

Kind regards,

Quentin Clairon, PhD

Guest Editor

PLOS Computational Biology

Hugues Berry

Section Editor

PLOS Computational Biology

**Additional Editor Comments:**

Dear Authors

All the comments of 2 out of the 3 reviewers have been positively answered to. Still, reviewer 1 asks for clarifications about the adequacy of your modifications and its comments.

Despite formally asking for major revision, this new step is more about clarification to see exactly where are your agreements and disagreements with reviewer 1.

**Reviewers' comments:**

Reviewer's Responses to Questions

**Comments to the Authors:**

Reviewer #1: Authors have provided a sequence of comments that follow my remarks stream. However, I request to include in that list the modifications done in the text. Besides, mark in some color the changes done in the main document. It is quite annoying to go back and forth trying to locate the changes (if any).

Reviewer #2: Thanks for the detailed answers to the first review and the work on updating the manuscript accordingly.

Reviewer #3: Line 812 " it is a subclass of the broader Markov models, where the gating particles may not be independent." This should be clarified, Markov models are inherently independent.

**Have the authors made all data and (if applicable) computational code underlying the findings in their manuscript fully available?**

The PLOS Data policy requires authors to make all data and code underlying the findings described in their manuscript fully available without restriction, with rare exception (please refer to the Data Availability Statement in the manuscript PDF file). The data and code should be provided as part of the manuscript or its supporting information, or deposited to a public repository. For example, in addition to summary statistics, the data points behind means, medians and variance measures should be available. If there are restrictions on publicly sharing data or code —e.g. participant privacy or use of data from a third party—those must be specified.requires authors to make all data and code underlying the findings described in their manuscript fully available without restriction, with rare exception (please refer to the Data Availability Statement in the manuscript PDF file). The data and code should be provided as part of the manuscript or its supporting information, or deposited to a public repository. For example, in addition to summary statistics, the data points behind means, medians and variance measures should be available. If there are restrictions on publicly sharing data or code —e.g. participant privacy or use of data from a third party—those must be specified.

Reviewer #1: **No:** These answers will depend of my comments done before.These answers will depend of my comments done before.

Reviewer #2: Yes

Reviewer #3: Yes

PLOS authors have the option to publish the peer review history of their article (what does this mean?). If published, this will include your full peer review and any attached files.). If published, this will include your full peer review and any attached files.

.

Reviewer #1: No

Reviewer #2: No

Reviewer #3: No

**Figure resubmission:**
---

## [Decision Letter · Decision Letter 2]

2 Apr 2026

Dear Mr Linkevicius,

We are pleased to inform you that your manuscript 'One model to rule them all: unification of voltage-gated potassium channel models via deep non-linear mixed effects modelling' has been provisionally accepted for publication in PLOS Computational Biology.

Best regards,

Quentin Clairon, PhD

Guest Editor

PLOS Computational Biology

Hugues Berry

Section Editor

PLOS Computational Biology

Reviewer's Responses to Questions

**Comments to the Authors:**

Reviewer #2: Thanks again for the detailed answers to the second review and the work on updating the manuscript accordingly as well as providing a detailed response and changelog.

Reviewer #3: none

**Have the authors made all data and (if applicable) computational code underlying the findings in their manuscript fully available?**

The PLOS Data policy requires authors to make all data and code underlying the findings described in their manuscript fully available without restriction, with rare exception (please refer to the Data Availability Statement in the manuscript PDF file). The data and code should be provided as part of the manuscript or its supporting information, or deposited to a public repository. For example, in addition to summary statistics, the data points behind means, medians and variance measures should be available. If there are restrictions on publicly sharing data or code —e.g. participant privacy or use of data from a third party—those must be specified.requires authors to make all data and code underlying the findings described in their manuscript fully available without restriction, with rare exception (please refer to the Data Availability Statement in the manuscript PDF file). The data and code should be provided as part of the manuscript or its supporting information, or deposited to a public repository. For example, in addition to summary statistics, the data points behind means, medians and variance measures should be available. If there are restrictions on publicly sharing data or code —e.g. participant privacy or use of data from a third party—those must be specified.

Reviewer #2: Yes

Reviewer #3: Yes

PLOS authors have the option to publish the peer review history of their article (what does this mean?). If published, this will include your full peer review and any attached files.). If published, this will include your full peer review and any attached files.

.

Reviewer #2: No

Reviewer #3: No

---

## [Editor Report · Acceptance letter]

PCOMPBIOL-D-25-00808R2

One model to rule them all: unification of voltage-gated potassium channel models via deep non-linear mixed effects modelling

Dear Dr Linkevicius,

I am pleased to inform you that your manuscript has been formally accepted for publication in PLOS Computational Biology. Your manuscript is now with our production department and you will be notified of the publication date in due course.

With kind regards,

Lilla Horvath
